# Identification of triacylglycerol remodeling mechanism to synthesize unusual fatty acid containing oils

Prasad Parchuri [1,4], Sajina Bhandari [1,4], Abdul Azeez [1], Grace Chen[2], Kumiko Johnson[2], Jay Shockey [3], Andrei Smertenko [1] & Philip D. Bates [1] ✉

Typical plant membranes and storage lipids are comprised of five common fatty acids yet over 450 unusual fatty acids accumulate in seed oils of various plant species. Plant oils are important human and animal nutrients, while some unusual fatty acids such as hydroxylated fatty acids (HFA) are used in the chemical industry (lubricants, paints, polymers, cosmetics, etc.). Most unusual fatty acids are extracted from non-agronomic crops leading to high production costs. Attempts to engineer HFA into crops are unsuccessful due to bottlenecks in the overlapping pathways of oil and membrane lipid synthesis where HFA are not compatible. *Physaria fendleri* naturally overcomes these bottlenecks through a triacylglycerol (TAG) remodeling mechanism where HFA are incorporated into TAG after initial synthesis. TAG remodeling involves a unique TAG lipase and two diacylglycerol acyltransferases (DGAT) that are selective for different stereochemical and acyl-containing species of diacylglycerol within a synthesis, partial degradation, and resynthesis cycle. The TAG lipase interacts with DGAT1, localizes to the endoplasmic reticulum (with the DGATs) and to puncta around the lipid droplet, likely forming a TAG remodeling metabolon near the lipid droplet-ER junction. Each characterized DGAT and TAG lipase can increase HFA accumulation in engineered seed oils.

Plant glycerolipids (including membrane lipids and storage oils) provide humans with food, essential nutrients, chemical feedstocks, and biofuels. The various uses/values of different plant lipids are predominately dependent on their fatty acid composition. Membrane lipids have a limited composition mostly composed of 16–18 carbon fatty acids with 0–3 double bonds. These common fatty acids are essential to maintaining membrane functions under diverse environmental conditions. However, storage oils (triacylglycerols, TAGs) found in seeds can accumulate a vast diversity of fatty acid structures with over 450 fatty acids characterized from different plant species[1]. Many of these "unusual" fatty acids could be valuable nutritional sources or renewable chemical feedstocks if high yields could be obtained through plant domestication, or via engineering unusual

fatty acid synthesis into a high-yielding oilseed crop[2–5]. Yet the limited success in engineering high levels of many unusual fatty acids in transgenic seed oils[6,7] indicates a need for enhanced understanding of the metabolic pathways involved in biosynthesis of diverse plant oil compositions.

Plant TAG is produced through a complicated multi-compartment metabolic network that overlaps with essential membrane lipid synthesis to produce diverse TAG molecular species[8–11]. Fatty acids are synthesized up to 18 carbons and 1 double bond in plastids, exported to the cytosol/endoplasmic reticulum (ER), and activated by esterification to coenzyme A. Acyl-CoA may be either utilized for glycerolipid assembly or further elongated to ≥20 carbons. The fatty acid composition of the acyl-CoA pool can be further modified through

[1]Institute of Biological Chemistry, Washington State University, Pullman, WA 99164, USA. [2]United States Department of Agriculture, Agricultural Research Service, Western Regional Research Center, Albany, CA 94710, USA. [3]United States Department of Agriculture, Agricultural Research Service, Southern Regional Research Center, New Orleans 70124 LA, USA. [4]These authors contributed equally: Prasad Parchuri, Sajina Bhandari. ✉e-mail: phil_bates@wsu.edu

"acyl editing" that exchanges acyl groups in/out of the membrane lipid phosphatidylcholine (PC), which is the extra-plastidial site for fatty acid desaturation in all plants, or other fatty acid modifications, such as hydroxylation, that occur in select plants. Plants assemble diacylglycerol (DAG), the immediate precursor to TAG, through two possible pathways. First, the classical Kennedy pathway involves ER assembly of phosphatidic acid from two acyl-CoAs and glycerol-3-phosphate with subsequent dephosphorylation for de novo biosynthesis of DAG. De novo DAG can also be used to produce ER membrane lipids such as PC. Second, removal of the PC phosphocholine headgroup produces PC-derived DAG. Because PC is the site for fatty acid modification, TAG produced from PC-derived DAG can have a different fatty acid composition than that produced from de novo DAG that relies on the acyl-CoA pool composition and Kennedy pathway acyltransferase selectivity. The final step in TAG synthesis is the transfer of a third fatty acid to DAG by either an acyl-CoA:diacylglycerol acyltransferase (DGAT) or phospholipid:diacylglycerol acyltransferase (PDAT). Thus, the path of fatty acid flux through the lipid metabolic network for acyl modification and the various acyltransferase selectivities for acyl donor and DAG molecular species both affect the final composition of TAG.

The PC-derived DAG pathway requires two-thirds of TAG fatty acids to move through the membrane lipid PC prior to TAG synthesis. For plants that accumulate nutritionally important polyunsaturated fatty acids through desaturation on PC (for example soybean, canola, *Arabidopsis thaliana*, *Camelina sativa*[12–15]), the PC-derived DAG pathway is an efficient mechanism for the flux of PC-modified fatty acids from PC to TAG. However, many unusual fatty acids are detrimental to membrane lipid structure/function[16], and thus the PC-derived DAG pathway could affect membrane biology during the synthesis of TAG. Indeed, the PC-derived DAG pathway represented a bottleneck to accumulation of the hydroxylated fatty acid (HFA), ricinoleic acid (18:1OH), in engineered Arabidopsis and Camelina[11,13,17–19]. This bottleneck is not present in castor beans (*Ricinus communis*) which produce ricinoleic acid on PC, and then utilizes acyl editing and Kennedy pathway de novo DAG to produce TAG containing three HFA (3HFA-TAG)[20,21], suggesting the Kennedy pathway may be optimal for accumulation of unusual fatty acids in some species.

Recent analysis of lipid metabolism in *Physaria fendleri* (formerly *Lesquerella fendleri*[22]) suggested a previously uncharacterized "TAG remodeling" pathway to overcome the PC-derived DAG bottleneck for accumulation of unusual fatty acids to 60% of seed oil[23]. *P. fendleri* is a Brassicaceae species closely related to Arabidopsis and Camelina but predominantly accumulates TAG molecular species containing two lesquerolic acids (20:1OH, produced from elongation of ricinoleoyl-CoA) at the *sn*-1 and *sn*-3 positions (2HFA-TAG). Lesquerolic acid is not present in PC, and thus a Kennedy pathway of DAG/TAG synthesis similar to castor that excludes HFA from membranes would be logical. However, transcriptomic analysis of developing *P. fendleri* seeds indicated most lipid biosynthetic genes were expressed similarly to other Brassicaceae species that utilize the PC-derived DAG pathway, including a key enzyme of PC-derived DAG synthesis phosphatidylcholine:diacylglycerol cholinephosphotransferase (PDCT)[24,25]. Therefore, an in vivo isotopic labeling approach was used to trace the lipid metabolic flux pathways in developing *P. fendleri* seeds[23]. The results indicated that *sn*-1,2 PC-derived DAG (without HFA) was initially utilized to synthesize 1HFA-TAG containing a single HFA at the *sn*-3 position. However, over time the 1HFA-TAG was converted to the major 2HFA-TAG molecular species with lesquerolic acid at both *sn*-1/*sn*-3. A TAG remodeling mechanism was proposed to explain the isotopic tracing where a lipase removes the *sn*-1 common fatty acid from initially produced 1HFA-TAG generating a *sn*-2,3-DAG molecule, and a second HFA is added to the *sn*-1 position, generating the final 2HFA-TAG. The proposed TAG remodeling solves the problem of how PC-derived DAG can be used to synthesize TAG containing *sn*-1 unusual fatty acids without first incorporating them into the membrane lipid

PC. However, the enzymes involved in this unique oil biosynthetic pathway were not identified[23].

Several key unknowns remain in the TAG remodeling hypothesis: (1) are the first and second TAG biosynthetic steps (at *sn*-3 then *sn*-1) catalyzed by a single DGAT or multiple enzymes? (2) can DGATs even use the *sn*-2,3-DAG isomer intermediate proposed in the pathway? (3) what type of lipase could selectively remove common fatty acids and not the HFA that accumulate? (4) TAG lipase activity during seed development leads to reductions in total oil in many species[26–29], how does the proposed TAG lipase in TAG remodeling not lead to TAG turnover? Here we utilized a series of molecular biology, genetic, cell biology, and biochemical approaches to answer these questions and identify the enzymes responsible for TAG remodeling. Our findings show that *P. fendleri* utilizes both PfeDGAT1 and PfeDGAT2 which have different selectivities for substrate acyl compositions and DAG stereochemical structures, and that a previously uncharacterized TAG lipase (PfeTAGL1) localizes to the ER and interacts with PfeDGAT1, forming a TAG remodeling metabolon. Additionally, we somewhat surprisingly demonstrate that a lipase alone (PfeTAGL1) can be utilized to engineer TAG remodeling into transgenic plants to accumulate enhanced levels of HFA in seed oils.

## Results

### *P. fendleri* acyl-CoA:diacylglycerol acyltransferases 1 and 2 have different specificities for substrate fatty acids and diacylglycerol stereochemical structures

To characterize enzymatic components of *P. fendleri* TAG remodeling pathway we reevaluated previous *P. fendleri* seed transcriptomic studies[24,25] to identify acyltransferase and TAG lipase candidates. Homologs of the genes encoding the structurally unrelated DGAT1, DGAT2, and DGAT3 enzymes[10,30] (Supplementary Fig. S1) were expressed in developing *P. fendleri* seeds with *PfeDGAT1* expressed 2-10 fold higher than *PfeDGAT2*, *PfeDGAT3*, or *PfePDAT1* across seed development, similar to *DGAT1* in other Brassicaceae species[25,31]. However, in developing castor seeds that accumulate 90% HFA in the oil, *RcDGAT2* is expressed ~80-fold higher than *RcDGAT1* and demonstrates higher selectivity for HFA-containing substrates[31–33]. PDAT1 utilizes PC as an acyl donor, but 20:1OH is not found in PC, indicating PfePDAT1 is not likely involved in 2HFA-TAG production. The role of DGAT3s in seed TAG accumulation remains unclear, with prior reports suggesting chloroplast localization and a role in TAG synthesis in vegetative tissues[34–36]. Thus, the expression analysis alone is insufficient to determining which class(es) of DGAT possesses the substrate and stereochemical selectivity required for TAG remodeling in *P. fendleri* embryos.

As a first screen to determine which of *P. fendleri* DGATs is a functional TAG synthesizing enzyme that can utilize HFA-containing substrates, we expressed *PfeDGAT1*, *PfeDGAT2*, and *PfeDGAT3* in the yeast quadruple mutant H1246 that does not produce TAG[37]. Both PfeDGAT1 and PfeDGAT2 could restore TAG biosynthesis but with much more TAG produced by PfeDGAT1 (Supplementary Fig. S2). *PfeDGAT3* expression failed to restore TAG biosynthesis. Similar results were found with yeast expressing cDNAs encoding *DGAT1*, *DGAT2*, and *DGAT3* enzymes from the related Brassicaceae species *Arabidopsis thaliana* as controls (Supplementary Fig. S2). Feeding hydrolyzed Physaria oil to the yeast-producing *P. fendleri* DGATs indicated that PfeDGAT1 could produce both 0HFA-TAG and 1HFA-TAG in vivo. PfeDGAT2 produced low amounts of 0HFA-TAG, but HFA-containing TAG species were undetectable (Supplementary Fig. S3). *PfeDGAT3* expressing yeast did not produce any TAG even in the presence of HFA substrates, thus further experiments focused on PfeDGAT1 and PfeDGAT2.

The yeast complementation experiments rely on host cell production and delivery of DAG and acyl-CoA molecular species to the transgenic DGAT enzymes; these metabolite profiles likely do not

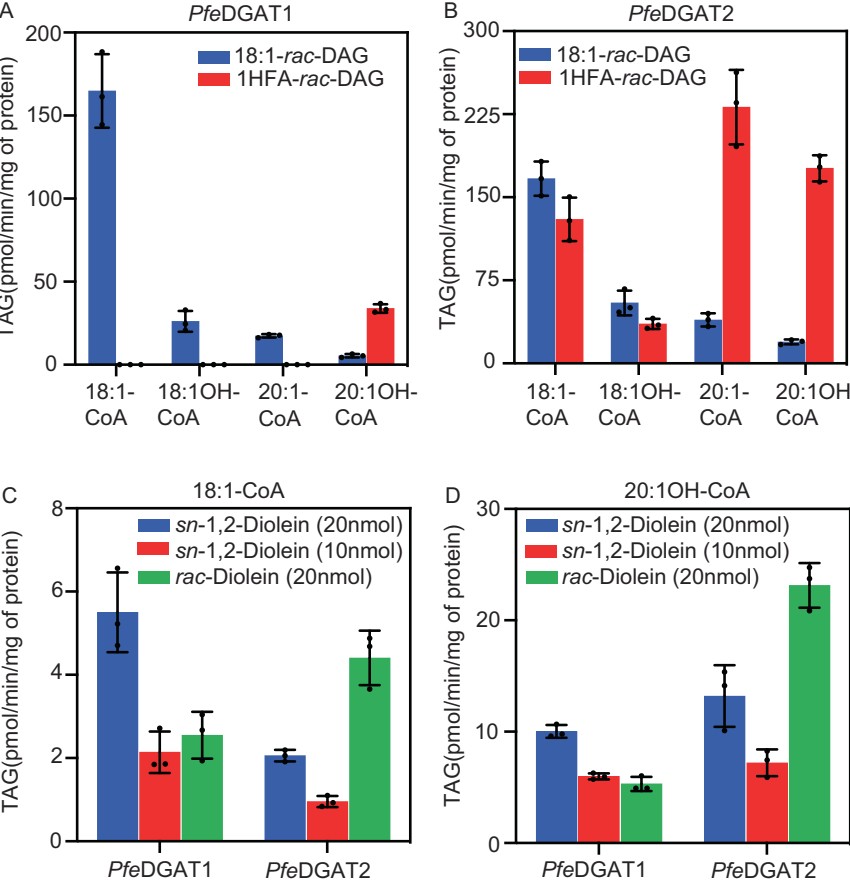

**Fig. 1 | Substrate specificities of *P. fendleri* DGATs. A, B** Acyl-CoA specificities of *PfeDGAT1* and *PfeDGAT2* with [14C]1,2-*rac*-diolein (20 nmol) and [14C]1HFA-*rac*-DAG (20 nmol) substrates. The [14C]1HFA-*rac*-DAG was derived from *P. fenderli* 2HFA-TAG and thus contains predominantly 20:1OH as the HFA. Acyl-CoA was added at a concentration of 5 nmol. **C, D** *sn*-1,2- and *sn*-2,3-DAG enantiomer specificities of

*PfeDGAT1* and *PfeDGAT2* with [14C]18:1-CoA (1 nmol) and [14C]20:1-OH-CoA (5 nmol). Activities are assayed using microsomes preparation from yeast expressing the enzymes. All the results are shown as mean ± SD of three independent assay reactions. Source data underlying **A**–**D** are provided in the source data file.

reflect the substrate molecular species availability in *P. fendleri* embryos. Synthesis of the major 2HFA-TAG molecular species in *P. fendleri* requires a DAG containing one 20:1OH as the acyl acceptor and 20:1OH-CoA as the acyl donor. Therefore, to investigate the substrate specificity of each *P. fendleri* DGAT for HFA-containing and non-HFA-containing molecular species of DAG and acyl-CoA, microsomes from *PfeDGAT1*- and *PfeDGAT2*-expressing yeast (Supplementary Figs. S3 and S4) were utilized for in vitro DGAT assays with various acyl-CoA and DAG substrates (Fig. 1). The [14C]1HFA-DAG substrate was produced through partial TAG lipase digestion of [14C]2HFA-TAG collected from [14C]acetate labeled developing *P. fendleri* seeds, which generates a racemic mixture of HFA-containing *sn*-1,2-DAG and *sn*-2,3-DAG species. A racemic mixture of [14C]dioleoyl-DAG representing a non-HFA-containing DAG molecular species was also used for each assay. The unlabeled 20:1OH-CoA acyl donor was synthesized from HFA collected from hydrolyzed *P. fendleri* 2HFA-TAG, and the unlabeled 18:1OH-, 20:1-, and 18:1- acyl-CoA were synthesized from purchased fatty acids. Each DGAT was assayed with each acyl-CoA and DAG molecular species combination. In contrast to the yeast feeding assays, the in vitro substrate specificity study revealed that PfeDGAT2 displayed higher total activity and produced more HFA-containing TAG molecular species than PfeDGAT1 (Fig. 1A, B), suggesting that differential fatty acid uptake or some other aspect of endogenous yeast metabolism limited access of PfeDGAT2 to HFA-containing substrates. In vitro both *P. fendleri* DGATs utilized all the acyl-CoA substrates provided to produce TAG, however, PfeDGAT1 was more active with the dioleoyl-DAG

substrate (Fig. 1A) whereas PfeDGAT2 was more active with the 1HFA-DAG substrate (Fig. 1B).

The TAG remodeling pathway hypothesizes two different stereochemical isomers of diacylglycerol are utilized for TAG synthesis, first *sn*-1,2-DAG derived from PC followed by *sn*-2,3-DAG (containing *sn*-3 HFA) derived from lipase action on 1HFA-TAG. DGAT stereoselectivity assays were designed to determine if PfeDGAT1 or PfeDGAT2 may be involved in the first, second, or both acyltransferase steps in the TAG remodeling pathway. The assays utilized unlabeled *sn*-1,2-diolein or a racemic (rac) diolein mixture (containing equal amounts of both *sn*-1,2 and *sn*-2,3 DAG) combined with either 14C-labeled 18:1-CoA (Fig. 1C) or 20:1OH-CoA (Fig. 1D) as the acyl donors. The results were similar regardless of the acyl donor species. PfeDGAT1 activity displayed half as much activity with 20 nmol rac-diolein as with 20 nmol of *sn*-1,2-diolein, indicating that PfeDGAT1 specifically acted upon the 10 nmol of *sn*-1,2-diolein isomer within the rac-diolein mixture. However, PfeDGAT2 could utilize both DAG stereochemical isomers, and had higher activity with the rac-diolein substrate indicating a preference for *sn*-2,3-DAG. Together the substrate molecular species and regiochemical specificity assays (Fig. 1), and prior developing seed gene expression data[24,25] suggest a TAG remodeling pathway where PfeDGAT1 (or PfeDGAT2) utilizes the non-HFA-containing *sn*-1,2 PC-derived DAG to produce the initial 1HFA-TAG, whereas only PfeDGAT2 can utilize the *sn*-2,3 HFA-DAG produced by lipase activity to synthesize the final 2HFA-TAG molecular species that accumulate in *P. fendleri* seeds.

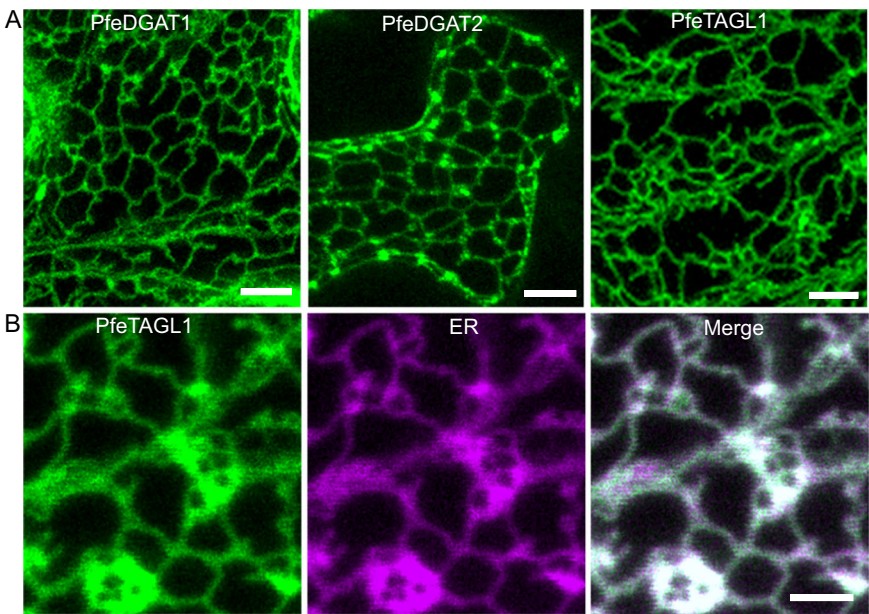

**Fig. 2 | Subcellular localization of *P. fendleri* DGATs and TAGLipase-like-1 in wild-type (WT) *N. benthamiana* leaf pavement cells. A** Confocal micrographs of transiently expressing GFP fusions of PfeDGAT1, PfeDGAT2, or PfeTAGL1. Scale bars = 10 μm. **B** Confocal micrographs of transiently expressing GFP fusion of PfeTAGL1 with ER (endoplasmic reticulum) maker fused to mCherry. Scale bar = 10 μm. Each experiment was repeated thrice independently with similar results.

## Subcellular localization and protein:protein interaction of TAG lipases with DGATs suggest a possible TAG remodeling metabolon

A central part of the TAG remodeling pathway in *P. fendleri* is an unknown lipase that removes the common fatty acid at *sn*-1 from 1HFA-TAG, producing *sn*-2,3-DAG with a HFA at *sn*-3. Developing *P. fendleri* seeds express multiple TAG lipase genes[25], including homologs of well-characterized TAG lipases involved in TAG turnover during seed maturation, seed germination, or pollen tube growth such as Sugar Dependent 1 (SDP1) and Oil Body Lipase 1 (OBL1)[38,39]. Remarkably, an uncharacterized TAG lipase gene (TAG lipase like-1; *PfeTAGL1*) was more highly expressed than any other candidate lipase gene during *P. fendleri* seed development[25]. The functions of TAGL1 remain unknown, though a proteomic analysis identified the *Arabidopsis thaliana* homolog AT1G23330 as a component of oil bodies[40]. SDP1 and OBL1 also function at oil bodies[41,42], whereas plant DGAT1 and DGAT2 reside in the ER. Differential localization of DGATs and TAG lipases presents a mechanistic dilemma on how TAG remodeling may occur. Interestingly, hydrophobicity plots indicate that PfeTAGL1 may contain a transmembrane domain for possible ER localization, unlike PfeSDP1 and PfeOBL1 (Supplementary Fig. S5). Therefore, to determine if Pfe-TAGL1 may also localize to the ER where TAG biosynthesis occurs, we created green fluorescent protein (GFP) fusions of PfeDGAT1, PfeD-GAT2, and PfeTAGL1 for subcellular localization experiments via confocal microscopy after transient expression in wild-type *Nicotiana benthamiana* leaf pavement cells. As expected both PfeDGAT1-GFP and PfeDGAT2-GFP localized to the ER (Fig. 2A). In agreement with our prediction, PfeTAGL1-GFP also localized to the ER (Fig. 2A). This finding was also further verified by successful ER co-localization of PfeTAGL1-GFP with a mCherry-ER marker[43] (Fig. 2B).

To examine possible localization of PfeDGAT1, PfeDGAT2, Pfe-TAGL1, PfeSDP1, and PfeOBL1 to oil bodies we utilized *N. benthamiana* leaves that have been previously engineered to form seed-like oil bodies in leaf pavement cells[44]. The oil bodies can be stained with Nile Blue for characterizing co-localization with GFP fusion proteins. Both DGAT-GFP fusions clearly associated with ER membranes and did not associate with oil bodies (Fig. 3A, B), while PfeSDP1 and PfeOBL1 were found to coat oil bodies but did not associate with ER (Fig. 3D, E).

PfeTAGL1 localized to the ER as in wild-type leaves and also labeled puncta surrounding oil bodies (Fig. 3C). It appears that oil bodies sequester PfeTAGL1 resulting in GFP signal enriched in the oil bodies relative to the ER. We measured the signal intensity of GFP and Nile Blue along the broken lines in Fig. 3A, C, and D; DGAT1 localized proximal to and partially overlapped with the oil body with a peak on one side of the oil body corresponding to ER (Fig. 3F). TAGL1 and OBL1 formed two peaks corresponding to the edges of the oil body with lower signal intensity in the center (Fig. 3G, H). The latter outcome is consistent with the conclusion that both proteins concentrate at the edges of the oil bodies.

Recent results have suggested that certain plant lipid assembly enzymes may associate as metabolons within the ER to support efficient utilization of select substrate pools (e.g. diacylglycerol) for the differential synthesis of membrane lipids or TAG[8,30,45,46]. Therefore, we investigated protein:protein interactions between the *P. fendleri* DGATs and lipases through split ubiquitin-based yeast two-hybrid (Y2H) and bimolecular fluorescence complementation (BiFC) (Fig. 4). Both approaches displayed complementary results. DGAT1 enzymes are known to function as dimers[30], and PfeDGAT1 interacted with itself (Fig. 4A, B, Ei). In addition, PfeDGAT1 interacted with PfeTAGL1 but not the other lipases (Fig. 4A, B, Eiii). The reconstituted PfeDGAT1-PfeTAGL1 BiFC signal localized in the ER where both proteins reside (Fig. 3, 4Eiii). PfeDGAT2 did not interact with any of the other enzymes. Together, the protein localization (Figs. 2 and 3) and protein:protein interaction (Fig. 4) results suggest that PfeTAGL1 localizes to the site of TAG biosynthesis and directly interacts with PfeDGAT1, thus making it a top candidate for involvement in the TAG remodeling pathway of 2HFA-TAG production in *P. fendleri*.

## PfeTAGL1 is a functional TAG lipase with a preference for common fatty acids over hydroxy fatty acids

*PfeTAGL1* was initially identified as a putative TAG lipase based on homology to other lipases[25], but the enzyme activity of PfeTAGL1 or any plant ortholog has yet to be determined. To confirm TAG lipase enzymatic activity *PfeTAGL1* was expressed in *S. cerevisiae* under the control of a galactose-inducible promoter, followed by tracking of lipid accumulation in induced cells, utilizing [$^{14}$C]acetate feeding.

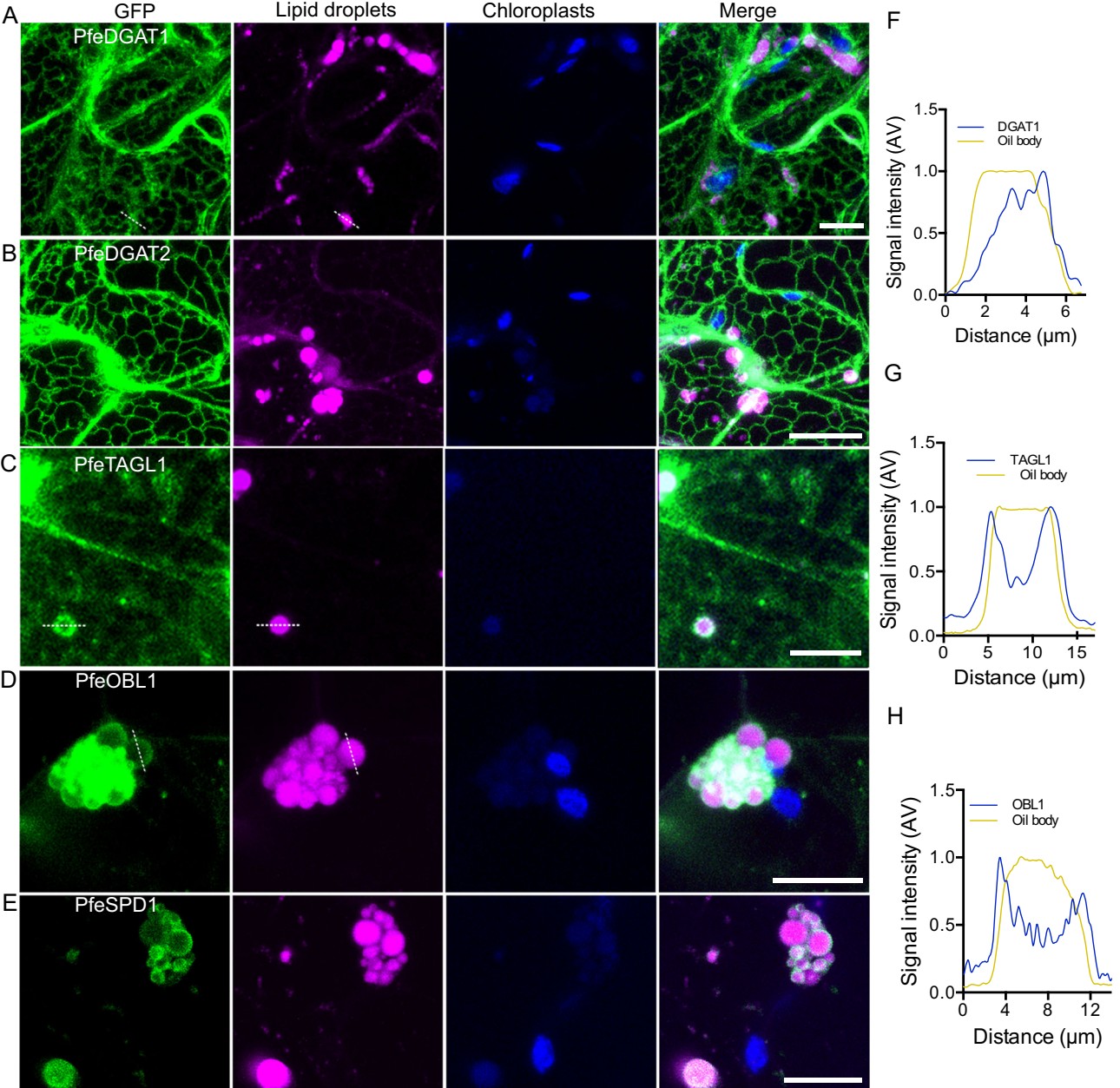

**Fig. 3 | Subcellular localization of *P. fendleri* DGATs and TAG lipases in High Oil (HO) *N. benthamiana* leaf pavement cells.** Confocal micrographs of transiently expressing GFP fusions of *P. fendleri* PfeDGAT1 (**A**), PfeDGAT2 (**B**), PfeTAGL1 (**C**), PfeOBL1 (**D**), and PfeSDP1 (**E**). Oil bodies were stained with 2 µg/ml Nile Blue A for 30 min. Scale bar = 10 µm. F-H, Normalized fluorescence intensity of GFP and Nile Blue A along the broken lines in panel **A** (**F**), panel **C** (**G**), and panel **D** (**H**). Each experiment was repeated thrice independently with similar results.

Compared to the vector control the *PfeTAGL1*-expressing yeast accumulated less TAG and more diacylglycerol and free fatty acids, the products of TAG lipase activity (Supplementary Fig. S6). For further enzymatic characterization, a His-tagged version of PfeTAGL1 was produced in *Escherichia coli* (Fig. 5A, B). Purified PfeTAGL1 displayed enzyme activity in vitro, unlike heat-inactivated PfeTAGL1, extracts from uninduced *E. coli*, and no enzyme controls (Fig. 5C). PfeTAGL1 had a pH optimum of 7 (Fig. 5D), and the release of free fatty acids from the TAG substrate was dependent on both time and protein concentration (Fig. 5E, F) consistent with enzymatic activity, thus confirming PfeTAGL1 as a functional TAG lipase.

The proposed TAG remodeling pathway suggests an unknown lipase may selectively remove the common fatty acid from the *sn*-1 position of 1HFA-TAG, generating an *sn*-2,3 DAG containing a *sn*-3 HFA[23]. Therefore, to test the acyl selectivity of PfeTAGL1 we performed a competitive TAG lipase assay in which the rate of [14C]triolein degradation is measured when mixed with different unlabeled TAG molecular species (Fig. 6A). The rate of [14C]triolein degradation by PfeTAGL1, measured as the loss of TAG and the appearance of free fatty acids, both increased in the presence of 1HFA-TAG and occurred even more in the presence of 2HFA-TAG isolated from *P. fendleri* seeds (Fig. 6B, C) indicating PfeTAGL1 preferentially cleaves common fatty acids from TAG rather than HFA. PfeSDP1 preferentially removed HFA from HFA-TAG[27] which supports the role of PfeSDP1 in 2HFA-TAG degradation, while PfeTAGL1 is involved in TAG remodeling to produce 2HFA-TAG.

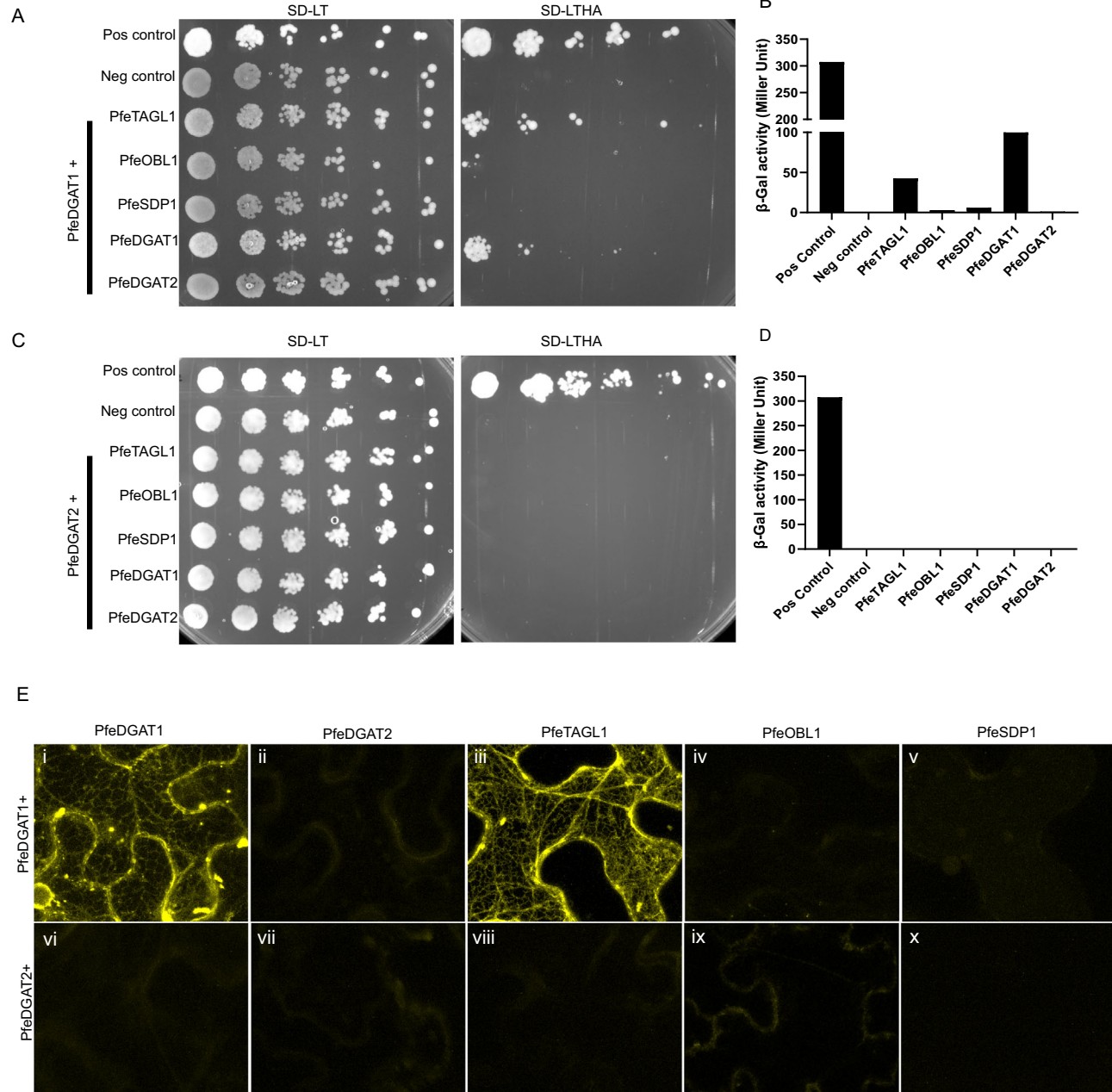

**Fig. 4 | Yeast two-hybrid assay and BiFC assay showing interaction of PfeDGATs with different *P. fendleri* prey genes. A** Prototypic growth assay of yeast containing bait pBT3C-PfeDGAT1 with various preys pPR3N-PfeTAG lipases and pPR3N-PfeDGATs on nonselective (SD-LT, left) and selective (SD-LTHA, right) media. Positive control yeast consisted of bait pCCW-Alg5 (Alg5-Cub-LexA) and prey pAI-Alg5 (Alg5-HA-NubI) and negative control had empty plasmids bait pBT3C and prey pPR3N. **B** Quantitative beta-galactosidase assay showing the strength of interaction of same bait and prey plasmid pairs. **C** Prototypic growth assay of yeast containing bait pBT3C-STE-PfeDGAT2 with various preys pPR3N-PfeTAG lipases and pPR3N-PfeDGATs on nonselective (SD-LT, left) and selective (SD-LTHA, right) media.

**D** Quantitative beta-galactosidase assay showing the strength of interaction of same bait and prey plasmid pairs. **E** Confocal images of WT *Nicotiana benthamiana* (i, ii, iii, vi, vii, viii) and HO *Nicotiana benthamiana* (iv, v, ix, x) leaves co-infiltrated with i. nYFPC-PfeDGAT1 and cYFPC-PfeDGAT1 ii. nYFPC-PfeDGAT1 and cYFPC-PfeDGAT2 iii. nYFPC-PfeDGAT1 and cYFPC-PfeTAGL1 iv. nYFPC-PfeDGAT1 and cYFPC-PfeOBL1 v. nYFPC-PfeDGAT1 and cYFPC-PfeSDP1 vi. nYFPC-PfeDGAT2 and cYFPC-PfeDGAT1 vii. nYFPC-PfeDGAT2 and cYFPC-PfeDGAT2 viii. nYFPC-PfeDGAT2 and cYFPC-PfeTAGL1 ix. nYFPC-PfeDGAT2 and cYFPC-PfeOBL1 x. nYFPC-PfeDGAT2 and cYFPC-PfeSDP1. Scale = 25 μm. Each experiment was repeated twice independently with similar results.

## RNAi suppression in *P. fendleri* indicates *DGAT1*, *DGAT2*, and *TAGL1* are all involved in accumulation of HFA-containing TAG

To investigate the in vivo roles of *PfeDGAT1*, *PfeDGAT2*, and *PfeTAGL1* on *P. fendleri* TAG biosynthesis seed-specific RNAi knockdown lines were produced (Fig. 7, Supplemental Figs. S7–9). Knockdown of each gene reduced total seed oil accumulation (Fig. 7A) and HFA content (Fig. 7B) confirming their role in HFA-containing TAG biosynthesis.

Importantly, this result confirms that both DGAT1 and DGAT2 are involved in *P. fendleri* TAG biosynthesis, whereas in Arabidopsis only the *dgat1* mutant has an oil phenotype but *dgat2* does not[47].

Knockdown of each gene was confirmed by qPCR, in addition we monitored compensating changes in *PfeDGAT1*, *PfeDGAT2*, *PfePDAT1*, and *PfeTAGL1* expression relative to wild-type in each line (Fig. 7C–E). In Arabidopsis, the *pdat1* mutant has no effect on seed oil, but AtPDAT1

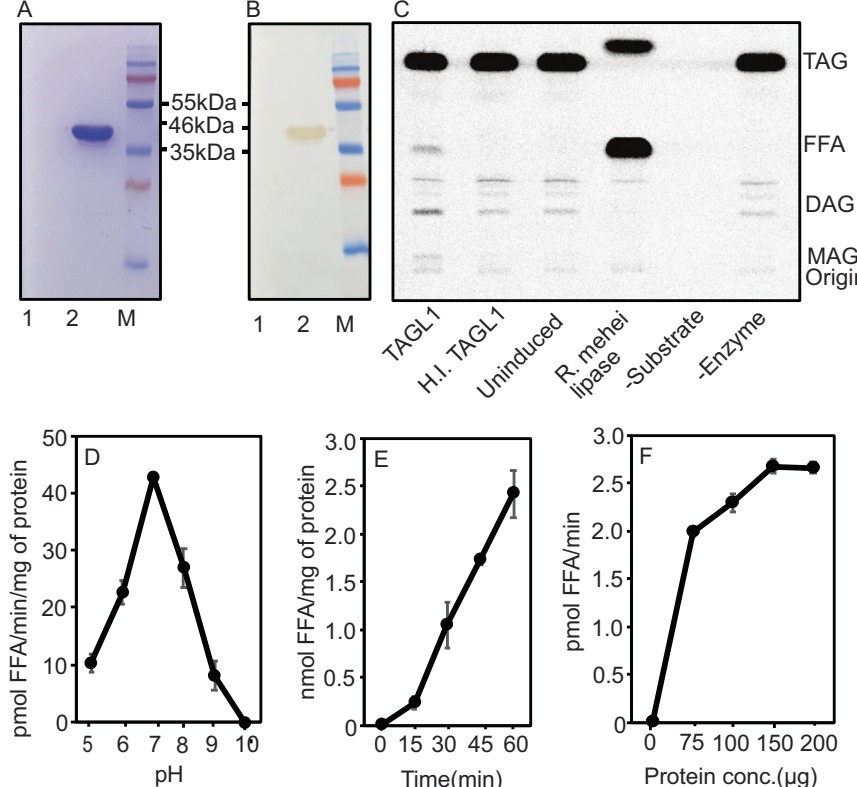

**Fig. 5 | *Physaria fendleri* triacylglycerol lipase-like 1 (TAGL1) encodes tria-cylglycerol lipase activity.** SDS-PAGE gel (**A**) and western blot (**B**) with anti-(His)$_{6x}$ antibody showing the purified His-tagged TAGL1 protein from *E. coli*. Lane 1, elution fraction from Uninduced cells; Lane 2, purified TAGL1 protein; and Lane M, marker. **C** TAG lipase activity of purified TAGL1 protein. Experiments **A**–**C** were repeated twice independently with similar results. Phosphor image TLC of in vitro lipase assay reactions with [$^{14}$C]triolein as substrate. Lipase from *Rhizopus mehei* was used

as positive control. Elution fraction from Induced cells, heat-inactivated (H.I.) TAGL1 and no enzyme addition were used as negative controls. The effect of pH (**D**), time (**E**), and protein concentration (**F**) on the rate of [$^{14}$C]triolein hydrolysis by TAGL1. The amount of substrate used was 5 nmol. All the results were expressed as mean ± SD of two independent experiments. Source data underlying **D**–**F** are provided in the source data file.

is responsible for TAG synthesis in the absence of AtDGAT1[47]. *PfePDAT1* expression was upregulated in both the *PfeDGAT1_RNAi* and *PfeD-GAT2_RNAi* lines likely partially compensating for reduced DGAT function. The increase in polyunsaturated fatty acids (rather than HFA) is also consistent with PC as an acyl donor for compensating PDAT activity in both *DGAT* knockdown lines (Fig. 7B). Interestingly, the *PfeDGAT1_RNAi* led to a larger increase in *PfeDGAT2* expression than *PfePDAT1*, however in the *PfeDGAT2_RNAi* line there was little to no increase in *PfeDGAT1* while *PfePDAT1* was greatly increased. The *DGAT1/2* expression results are consistent with their relative enzymatic activities. PfeDGAT2 can utilize both *sn*-1,2-DAG and *sn*-2,3-DAG and thus can compensate for the loss of PfeDGAT1. However, PfeDGAT1 can only use *sn*-1,2-DAG and thus cannot replace PfeDGAT2 for the use of *sn*-2,3-DAG during TAG remodeling (Fig. 1C, D). The large compensating increase in *PfePDAT1* in the *PfeDGAT2_RNAi* background may suggest that PfePDAT1 can also utilize *sn*-2,3 DAG.

Reduction of *PfeTAGL1* expression led to a large increase in 18:1 content, and the largest decrease in HFA (Fig. 7B) suggesting that 18:1 normally released by PfeTAGL1 during TAG remodeling is a substrate for further HFA production, consistent with the in vivo labeling[23]. In *PfeTAGL1_RNAi*, *PfeDGAT2* expression greatly decreased and *PfePDAT1* expression did not increase, therefore without the production of *sn*-2,3 DAG by PfeTAGL1 PfeDGAT2 activity may not be needed. However, *PfeDGAT1* expression did increase suggesting one consequence of a block in TAG remodeling is an attempt to increase flux through the early part of the pathway. Together the in vivo knockdown results are consistent with each enzyme's in vitro activities and roles in TAG remodeling.

## *P. fendleri* DGATs and TAGL1 can enhance the engineering of HFA accumulation in Arabidopsis, a plant that does not naturally produce HFA

The *in planta* functions of PfeDGAT1, PfeDGAT2, and PfeTAGL1 were further evaluated by expression of the respective cDNAs in Arabidopsis seeds that had been previously engineered to produce HFA by overexpression of the castor fatty acid hydroxylase *RcFAH12* gene (Fig. 8, Supplementary Fig. S12A–D). The background RcFAH line predominantly produces 18-carbon HFAs which accumulate at the *sn*-2 position of TAG[48], but also produces low levels of the 20-carbon HFAs found in *P. fendleri* (Fig. 8C). Previous results in HFA-producing Arabidopsis have indicated that inefficient HFA accumulation in TAG induces feedback inhibition of fatty acid synthesis, thus lowering total seed oil levels[13,17]. Expression of both *PfeDGAT1* and *PfeDGAT2* in HFA-Arabidopsis seeds increased total seed oil accumulation (Fig. 8A) and total HFA content (Fig. 8B, C, Supplementary Fig. S12A, B), but PfeD-GAT1 was more effective than PfeDGAT2. Expression of *PfeTAGL1* in the RcFAH background created the largest increases in total seed oil and HFA accumulation (Fig. 8A–C and Supplementary Fig. S12C). Considering that TAG lipases have been characterized as involved in TAG breakdown, not TAG synthesis, this result was a surprise. To compare, expression of *PfeSDP1* (the major enzyme involved in oilseed TAG turnover[27,49]) in WT or HFA-producing Arabidopsis seeds led to reductions in total seed oil accumulation (Supplementary Fig. S10), thus further emphasizing a distinct role for PfeTAGL1 compared to other characterized oilseed TAG lipases[39].

Expression of each *P. fendleri* gene in Arabidopsis had a unique effect on the whole seed fatty acid composition (Fig. 8C) representing

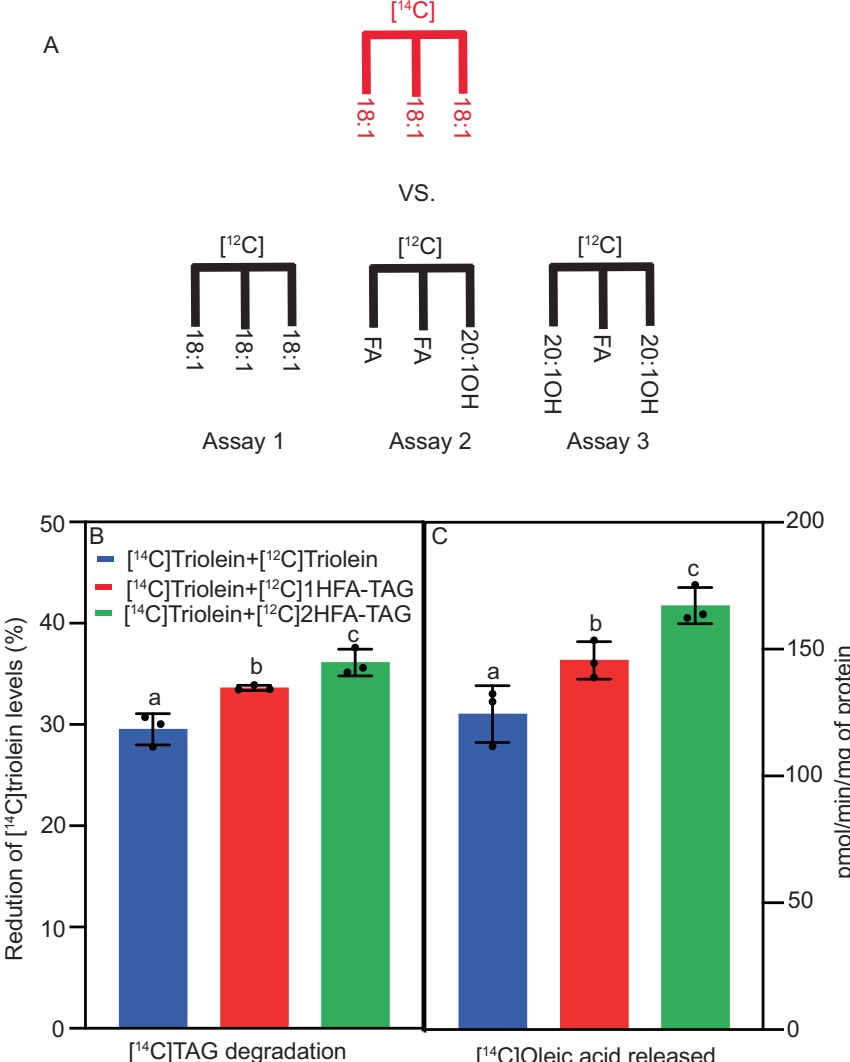

**Fig. 6 | Fatty acid selectivity of *Physaria fendleri* TAGL1. A** Competitive lipase assay model. Equimolar amounts (1:1) of [14]C-labeled triolein along with either unlabeled triolein (Assay 1) or 1HFA-TAG (Assay 2) or 2HFA-TAG (Assay 3) were used as a substrates. In assays 2–3, if a lipase selectively cleaves oleate over other fatty acids the rate of [14]C]triolein degradation will increase compared to 1. However, if the lipase prefers other fatty acids over oleate the rate of [14]C]triolein degradation will decrease. Lipase activity of PfeTAGL1 protein was represented as the amount (%) of [14]C]triolein degraded (**B**) and release of [14]C]oleic acid (**C**) as measured by phosphor imaging. Results are shown as Mean ± SD of three independent assay reactions. Means with different letters are significantly different according to the one-way ANOVA with LSD test at $p < 0.05$. Source data underlying **B**, **C** and exact $p$ values are provided in the source data file.

the differential functions and substrate selectivities of each enzyme. Additionally, TAG molecular species were separated based on HFA content and the fatty acid composition determined (Fig. 9). In each molecular species *PfeDGAT1* expression led to the highest levels of oleic acid (18:1) consistent with its in vitro preference for DAG containing 18:1 (Figs. 9 and 1A). In the HFA-containing TAGs (Fig. 9B, C) PfeDGAT1 activity enhanced densipolic acid (18:2OH) accumulation at the expense of ricinoleic acid (18:1OH) and lesquerolic acid (20:1OH). PfeDGAT2 activity led to the highest levels of 20:1 and 20:1OH in all TAG fractions, especially in 2HFA-TAG (Fig. 9C) which is consistent with its in vitro preference for utilizing both DAG and acyl-CoA containing 20-carbon fatty acids, particularly 20:1OH (Fig. 1B). Expression of PfeTAGL1 uniquely increased polyunsaturated fatty acids (PUFA: 18:2, 18:3, 18:2OH) in all TAG molecular species. Unlike PfeDGAT1 and PfeDGAT2, PfeTAGL1 does not synthesize TAG, thus the changes in the TAG molecular species compositions are reflective of the lipase activity (likely on both 1HFA-TAG & 0HFA-TAG produced by Arabidopsis) combined with the fate of released fatty acids that are

incorporated back into glycerolipid biosynthesis. The increase in PUFA suggests that 18:1/18:1OH released by PfeTAGL1 activity re-entered the PC pool for additional desaturation. The previous isotopic tracing of *P. fendleri* lipid metabolism also measured increases in polyunsaturated fatty acids during TAG remodeling[23] which is consistent with the function of PfeTAGL1 on Arabidopsis lipid metabolism.

## Discussion

### TAG remodeling is controlled by acyl- and regiochemical-selective DGATs and an unique interacting TAG lipase

The control of seed oil fatty acid composition by TAG remodeling was first suggested through in vivo isotopic tracing of lipid metabolism in *P. fendleri*[23], yet the enzymatic mechanisms of TAG remodeling were unknown. Here we present experimental evidence and a model (Fig. 10) for the enzymatic mechanisms of TAG remodeling in *P. fendleri* seed oil biosynthesis.

Like other Brassicaceae plants *P. fendleri* utilizes PC-derived *sn*-1,2-DAG containing common fatty acids to synthesize TAGs[12,13,23]. In

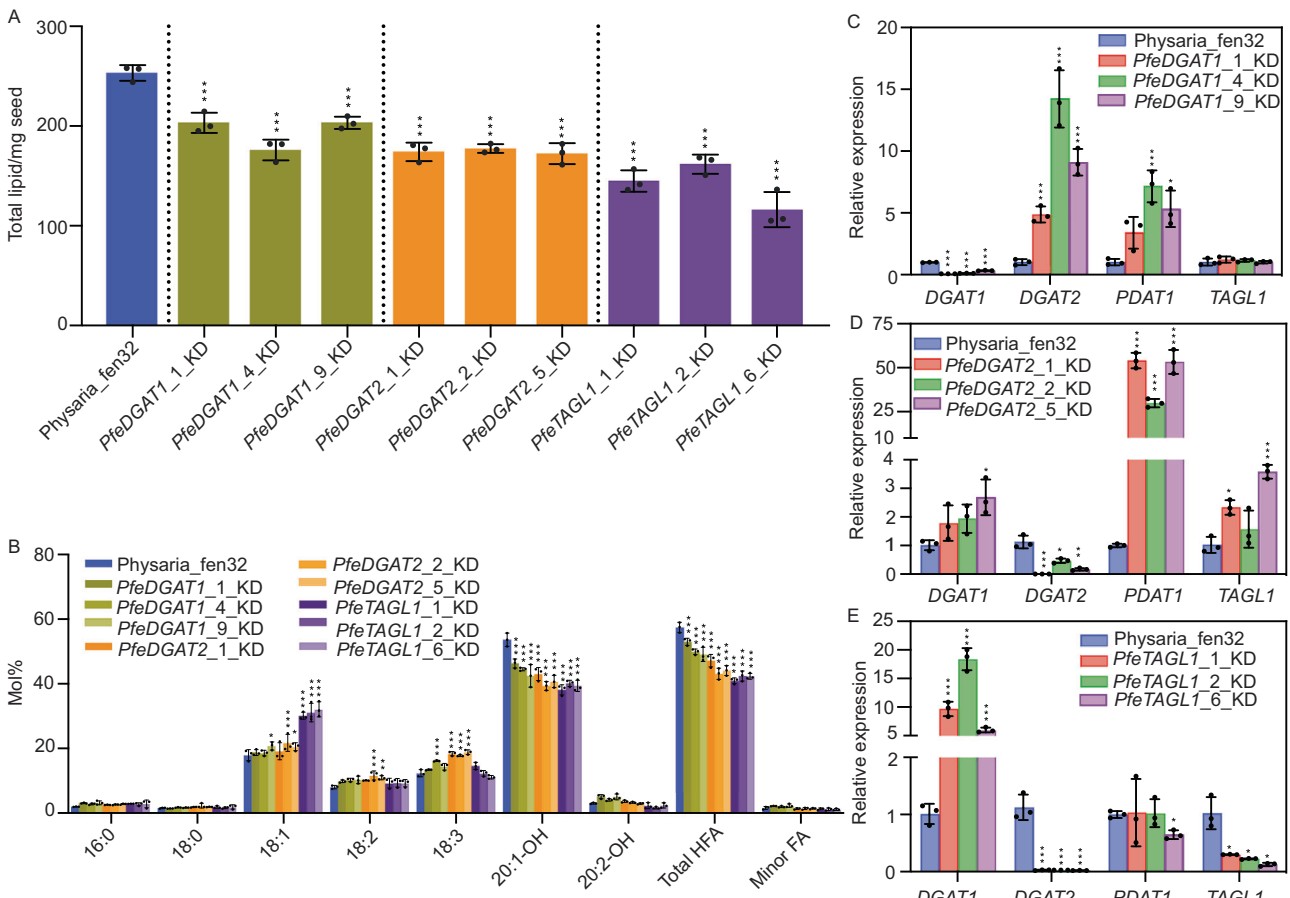

**Fig. 7 | Knockdown of TAG remodeling genes (*Pfe*DGAT1, *Pfe*DGAT2, and *Pfe*-TAGL1) in Physaria. A** Total lipid content of T2 seeds of selected lines. Data represents mean ± SD (*n* = 3) of three biological replicates. Lines significantly different (*p* < 0.05) than Physraia_fen32 determined by Ordinary one-way ANOVA or with Dunnett's multiple comparisons correction are marked with a black asterisk. **B** Fatty acid composition of T2 seeds of selected lines. Physaria_fen32 was used as wild-type control. Data represents mean ± SD (*n* = 3) of replicates. Lines significantly different (*p* < 0.05) than Physraia_fen32 determined by two-way ANOVA with Dunnett's multiple comparisons correction are marked with a black asterisk. See Supplementary Figs. S8 and S9 for the data of all the knockdown lines for each gene. Relative expression of *Pfe*DGAT1, *Pfe*DGAT2, *Pfe*PDAT1, and *Pfe*TAGL1 genes in selected *Pfe*DGAT1 knockdown lines (**C**), PfeDGAT2 knockdown lines (**D**), Pfe-TAGL1 knockdown lines (**E**). The gene expression was quantified by qRT-PCR and normalized to 18S internal reference gene. Physaria_Fen 32 was used as calibrator. Data represent the mean ± SD of three biological replicates. A significant difference between control and knockdown lines was determined using a multiple *t*-test with the Holm-Sidak method and a confidence interval of 95% (*$p \le 0.33$, **$p \le 0.002$ and ***$p \le 0.001$). Source data underlying **A**–**E** and exact *p* values are provided in the source data file.

*P. fendleri* the *sn*-3 acylation of PC-derived DAG with 20:1OH produces a 1HFA-TAG molecular species. However, during *P. fendleri* TAG remodeling, the initial 1HFA-TAG product (containing *sn*-3 HFA) is cleaved to *sn*-2,3-DAG (by releasing the *sn*-1 common fatty acid) before a second HFA is added to the *sn*-1 position, producing the final 2HFA-TAG molecular species (Fig. 10). The more highly expressed PfeDGAT1[24,25] only utilized the *sn*-1,2 DAG isomer, and was more selective for DAG species containing common fatty acids than HFA, whereas PfeDGAT2 was highly active with HFA-containing substrates and could utilize *sn*-2,3-DAG (Fig. 1). Therefore, PfeDGAT1 and PfeD-GAT2 likely perform the first and second DGAT reactions, respectively (Fig. 10). The results of the RNAi knockdown lines and compensating gene expression changes (Fig. 7) is consistent with this conclusion. The proposed use of different DAG pools by PfeDGAT1 and PfeDGAT2 is also consistent with reports of tung tree DGAT1 and DGAT2 localizing to different subdomains of the ER membrane[50]. The current findings are also supported by isotopic tracing in transgenic Arabidopsis that indicated AtDGAT1 efficiently utilized initially produced PC-derived DAG, whereas transgenic expression of DGAT2s from Arabidopsis, soybean, and castor all utilized a distinct and more slowly metabolized PC-derived DAG pool[45].

The key aspect of TAG remodeling in *P. fendleri* is the lipase that removes the *sn*-1 common fatty acid from 1HFA-TAG, generating *sn*-2,3-DAG that can be utilized by PfeDGAT2 to produce 2HFA-TAG. Because TAG remodeling is part of the *P. fendleri* oil biosynthetic pathway, the lipase involved must not lead to TAG and HFA turnover as has been characterized for other TAG lipases[27,39]. Here we report that PfeTAGL1 is responsible for TAG remodeling based on: expression in developing seeds;[25] localization to the ER and puncta around oil bodies that could represent ER-oil body junctions where TAG biosynthesis occurs (unlike TAG degrading lipases which localize to the oil body (Figs. 2 and 3)); direct interaction with PfeDGAT1, which would facilitate possible substrate channeling (Fig. 4); a TAG lipase activity selective for common fatty acids over HFA (Figs. 5 and 6); *Pfe*-*TAGL1_RNAi* knockdown lines that display oil amounts and fatty acid composition consistent with a role in TAG remodeling (Fig. 7); and whose transgenic expression leads to increases in HFA-containing TAG (Fig. 8) rather than TAG degradation, as seen with expression of *PfeSDP1* (Supplementary Fig. S10). Therefore, the steps responsible for producing 2HFA-TAG containing HFA at the *sn*-1,3 positions starting from PC-derived DAG that does not contain HFA in *P. fendleri* are the consecutive action of PfeDGAT1, PfeTAGL1, and PfeDGAT2 (Fig. 10).

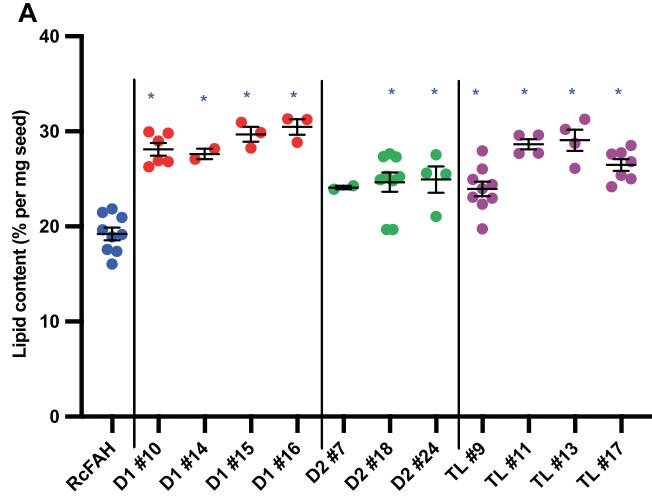

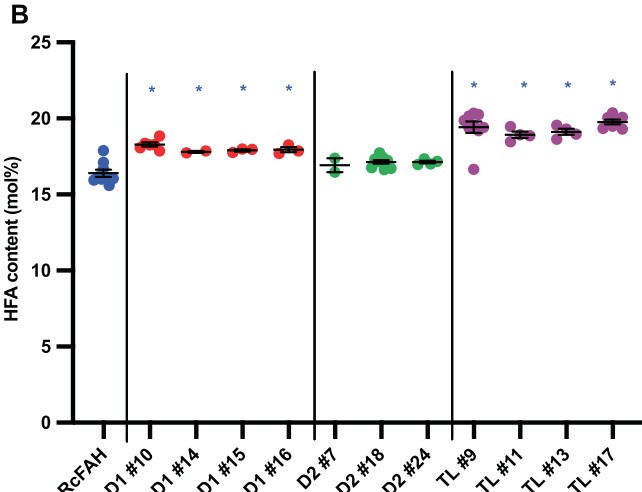

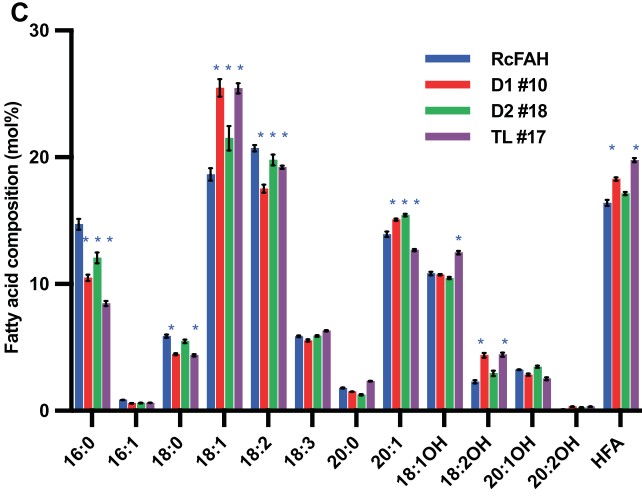

**Fig. 8 | Seed fatty acid content and composition of Arabidopsis engineered for TAG remodeling.** T3 seeds expressing *P. fendleri* genes *Pfe*DGAT1 (D1-1), *Pfe*DGAT2 (D2) and *Pfe*TAGLipase-like-1 (TL) in *Arabidopsis thaliana* RcFAH line. **A** Total lipid content. Data represents mean ± SEM of individual lines ($n = 9$ for RcFAH, D2#18, and TL#9; $n = 7$ for TL#7; $n = 6$ for D1#10; $n = 4$ for D2#24, TL#11 and TL#13; $n = 3$ for D1#15 and D1#16; $n = 2$ for D1#14 and D2#7. Lines significantly different ($p < 0.05$) than RcFAH determined by Ordinary one-way ANOVA with Dunnett's multiple comparisons correction are marked with a blue asterisk. **B** HFA content. Data represents mean ± SEM individual lines ($n$ value is the same as above in **A**). Lines significantly different ($p < 0.05$) than RcFAH determined by Ordinary one-way ANOVA with Dunnett's multiple comparisons correction are marked with a blue asterisk. **C** Fatty acid composition of T3 lines selected for further analysis. Data represents mean ± SEM individual lines (n value is the same as above in A). Lines significantly different ($p < 0.05$) than RcFAH determined by two-way ANOVA with Dunnett's multiple comparisons correction are marked with a blue asterisk. Source data underlying **A**–**C** and exact p values are provided in the source data file.

compositions through two mechanisms: (A) direct unusual fatty acids into the Kennedy pathway as in castor beans or Cuphea sp[20,52,53]. or (B) through the remodeling of TAG initially produced from PC-derived DAG not containing unusual fatty acids[23] by the utilization of acyl and stereoselective TAG lipase and DGATs as demonstrated here in *P. fendleri* (Fig. 10).

The enzyme localization and protein interaction results presented here (Figs. 2–4) and portrayed in Fig. 10 are consistent with the recent hypotheses that plant lipid metabolism in the ER is arranged into metabolons for the organization and control of lipid metabolic flux[8,45,46]. This metabolic organization allows multiple discreet pools of the same lipid class (e.g. DAG) to exist in the ER, and to be differentially utilized to produce select lipid molecular species. Thus, the spatial separation of de novo DAG biosynthesis (that is used to produce membrane lipids such as PC) into a different metabolon than TAG biosynthesis that utilizes PC-derived DAG likely explains why *P. fendleri* cannot produce 2HFA-TAG through a traditional Kennedy pathway. Hydroxy fatty acids are incompatible with membrane lipid structure/function[16], therefore TAG remodeling may have evolved as a way to circumvent the bottleneck of unusual fatty acids flux through PC for the PC-derived DAG pathway of TAG biosynthesis. The differential localization of TAG lipases here, and the interaction of PfeTAGL1 with PfeDGAT1 also support the metabolon hypothesis and demonstrate how the cellular context is key to understanding the function of previously uncharacterized proteins. The metabolon hypothesis and the DGAT DAG substrate selectivities (Fig. 1) may also help to explain why *PfeDGAT1* and *PfeDGAT2* are expressed at similar levels in *P. fendleri* seeds[24,25]. Each enzyme is required and utilizes a different DAG pool for producing different TAG molecular species (Fig. 10).

**Possibility of TAG remodeling in other oil-accumulating plants**

*P. fendleri* was the first species identified to use TAG remodeling as a major pathway of seed oil production[23]. Yet, there are hints in the literature that TAG remodeling may be required in other species as well. For example, *Crambe abyssinica* accumulates ~60% erucic acid (22:1) in oil which accumulates at the *sn*-1,3 positions of TAG and little to none in PC[54,55], similar to 20:1OH in *P. fendleri*. In vivo isotopic tracing of lipid metabolism and RNAi silencing of *PDCT* both support the use of PC-derived DAG for TAG biosynthesis[56–58], and GPAT assays indicate *C. abyssinica* de novo DAG synthesis would not utilize 22:1-CoA[57]. Thus, how 22:1 is incorporated into the *sn*-1 position of *C. abyssinica* TAG by either de novo DAG or PC-derived DAG is unclear, but TAG remodeling to incorporate *sn*-1 erucic acid could be a viable explanation. While time-dependent isotopic tracing of TAG molecular species synthesis is the gold standard for measuring TAG remodeling[23], the characterization of PfeTAGL1 as involved in ER-localized TAG remodeling, combined with the differential DAG substrate selectivities of PfeDGAT1 and PfeDGAT2, indicate that relatively high expression of

**Beyond *P. fendleri:* insights on plant lipid metabolism**

Plants contain two main pathways to produce the DAG utilized for seed TAG biosynthesis, de novo DAG biosynthesis through the Kennedy pathway and PC-derived DAG biosynthesis[8]. Despite the prevalence of PC-derived DAG biosynthesis in many oilseed plant species[11–14,23], the PC-derived DAG pathway of TAG biosynthesis presents a metabolic bottleneck for plants that accumulate unusual fatty acids[13,16,17,51]. However, plants have evolved ways to efficiently accumulate unusual fatty acids in TAG while maintaining proper membrane lipid

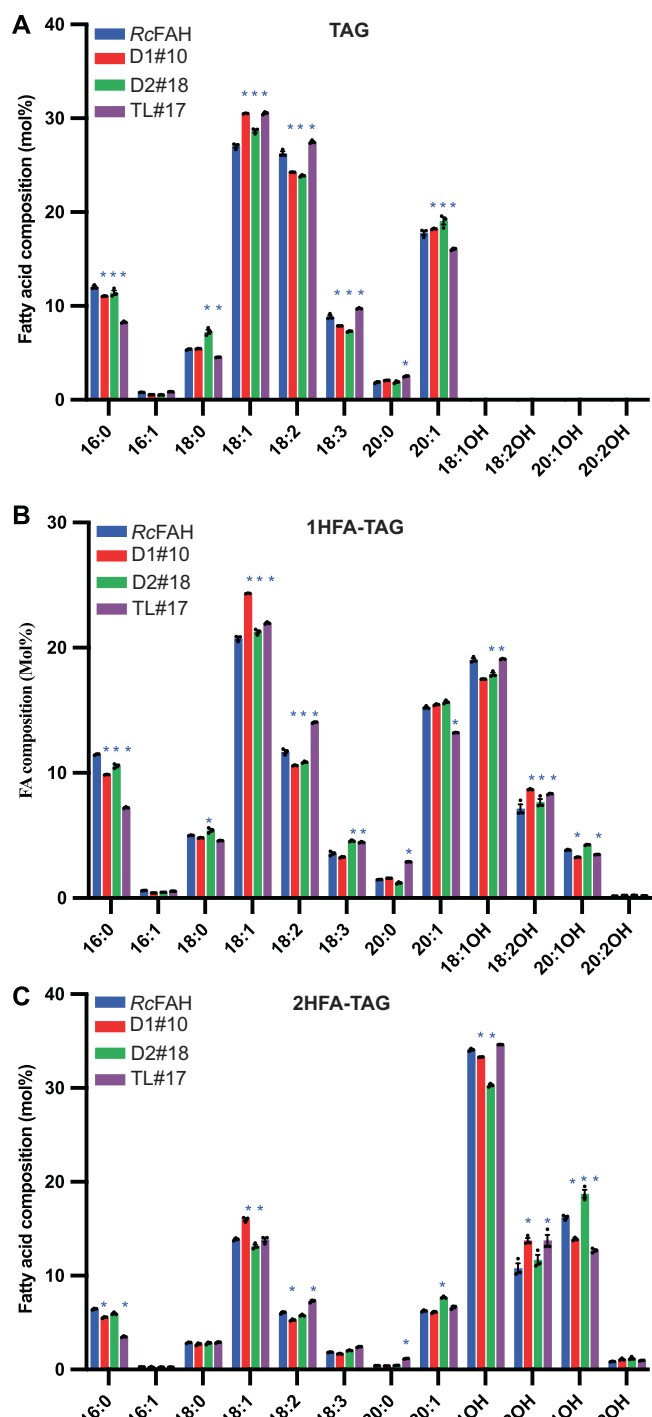

**Fig. 9 | Fatty acid composition of different TAG molecular species in T4 Arabidopsis seeds expressing *P. fendleri* genes.** *Pfe*DGAT1 (D1), *Pfe*DGAT2 (D2), and *Pfe*TAGLipase-like-1 (TL) in *Arabidopsis thaliana* RcFAH line. **A** TAG. **B** 1HFA-TAG. **C** 2HFA-TAG. Data are mean ± SEM of three biological replicates for each line. Significant differences compared to control (RcFAH) are determined by two-way ANOVA and marked with a blue asterisk ($p$ value < 0.05). Source data underlying **A**–**C** and exact $p$ values are provided in the source data file.

the three orthologous enzymes in other developing oilseeds may be indicators of the presence of TAG remodeling. However, as of yet, it is unclear if DGAT1s and DGAT2s from other species also have differential selectivity for DAG stereochemical structures, thus further characterization of the enzymatic selectivities of DGAT enzymes

across the kingdoms of life will aid in understanding the roles of these distinct enzymes in TAG metabolism.

### Engineering TAG remodeling to control plant oil fatty acid composition

The identification of enzymes responsible for *P. fendleri* TAG remodeling presents opportunities to use TAG remodeling as an engineering strategy to control oilseed crop fatty acid compositions. Overexpression of both *PfeDGAT1* and *PfeDGAT2* increased total seed oil and HFA content (Figs. 8, 9, Supplementary Fig. S12), similar to HFA-selective acyltransferases from castor[7] further supporting the observation that efficient accumulation of HFA into seed oils can alleviate HFA-induced inhibition of fatty acid synthesis in engineered plants[17]. Interestingly, in both yeast (Supplemental Figs. 1 and 2) and Arabidopsis (Figs. 8 and 9) *PfeDGAT1* expression led to larger increases in total TAG and HFA while in vitro PfeDGAT2 had the highest activity, and in both in vitro and *in planta* PfeDGAT2 produced the most 2HFA-TAG containing 20:1OH (Figs. 1B and 9C). The differential effect of PfeDGAT1 and PfeDGAT2 on TAG content in heterologous systems is likely related to both substrate selectivity of the DGATs and substrate availability. Both yeast and Arabidopsis predominantly produce *sn*-1,2-DAG for TAG synthesis and even in the engineered (or HFA-fed) systems, this DAG pool will contain minimal 20:1OH (Fig. 8) and likely not contain *sn*-1 HFA[13]. Therefore, the substrate availability in these systems likely does not favor PfeDGAT2. Thus, the utilization of PfeDGAT2 (and possibly other *sn*-2,3 DAG selective DGATs) for oilseed engineering will likely require the production of *sn*-2,3 DAG containing the preferred fatty acids through the combined expression of a TAG lipase such as PfeTAGL1. The fact that *PfeTAGL1* expression in transgenic HFA-Arabidopsis seeds increased total oil and HFA content alone (Fig. 8), suggests that the increased *sn*-1,2-/-2,3 DAG production from degradation of the endogenous 0HFA-TAG provided more DAG substrate for enhanced HFA incorporation into TAG by the endogenous acyltransferases, thereby alleviating the HFA-mediated inhibition of de novo fatty acid synthesis[17]. Beyond HFA engineering, the increased PUFA produced by *PfeTAGL1* expression indicates that TAG lipases alone can be utilized to modify seed oil compositions. Lipases such as PfeTAGL1 that have acyl selectivity, interact with a DGAT, and localize to the ER are likely beneficial to engineer TAG remodeling without a reduction in seed oil content. Therefore, the induction of TAG remodeling by the combined expression of acyl-selective TAG lipases and DGATs during seed oil biosynthesis may be a valuable approach to engineer seed oil fatty acid composition after initial synthesis by the host plant.

## Methods
### Chemicals and substrates
All chemicals are from Fisher Scientific unless stated otherwise. [14C]-labeled acyl-CoA (18:1-CoA) and Triolein (Carboxyl-[14C]) are from American Radiolabeled Chemical, Inc., United States. [14C]2HFA-TAG substrate was purified using TLC (Analtech Silica gel HL; 20 cm × 20 cm; 250 μm thickness; 15 μm particle size) from total lipids extracts of *P. fendleri* developing embryos which are cultured in the presence of [14C]acetate as in[23]. 1,2-*sn*-Diolein was purchased from Avanti Polar Lipids, United States. [14C]-labeled *sn*-1,2/2,3-rac-diolein and *sn*-1,2/2,3-rac-1HFA-DAG was prepared from [14C]Triolein and [14C]2HFA-TAG, respectively by partial lipase treatment using *Rhizomucor mehei* TAG lipase (Sigma-Aldrich). TAG digestion, TLC separation, and production elution were done as in ref. 59, except the [14C]2HFA-TAG digestion products were separated with the double development system chloroform: methanol: acetic acid (97:3:0.5 and 99:0.5:0.5, v/v) as in ref. 13. Unlabeled [12C]1HFA-TAG and 2HFA-TAG were purified using TLC from Physaria seed oil as per ref. 23. Non-radiolabeled fatty acids, 18:1, 18:1-OH, 20:1 were obtained from Nu-Chek Prep, Inc., United States. [14C]20:1-OH and [12C]20:1-OH fatty acids was purified from [14C]

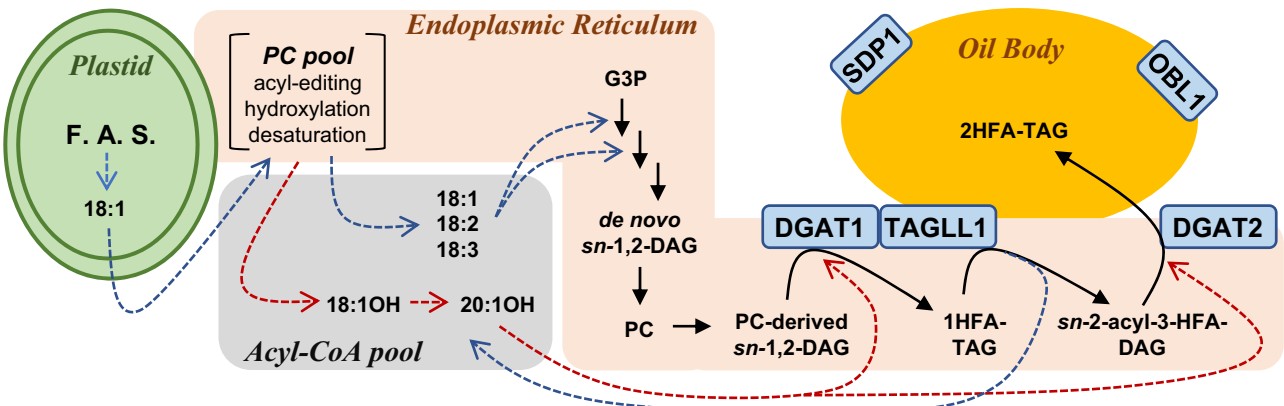

**Fig. 10 | Organization of TAG biosynthesis through TAG remodeling in *P. fendleri*.** Dashed arrows are acyl transfers, blue are common fatty acids, and red is hydroxy fatty acids. Solid lines involve glycerol backbone flux. Enzymes characterized by activity, interaction, and localization in this study are in blue boxes. The model utilizes at least two PC pools, one involved in acyl editing and fatty acid hydroxylation/desaturation. The second pool of PC is involved in PC-derived DAG biosynthesis containing common fatty acids. The model also involves at least three pools of DAG: (1) de novo *sn*-1,2-DAG produced by the Kennedy pathway utilized for PC synthesis; (2) PC-derived *sn*-1,2-DAG utilized by DGAT1 to produce 1HFA-TAG containing the HFA at the *sn*-3 position; (3) TAGLL1 activity on 1HFA-TAG produces a *sn*-2,3-DAG with the HFA at the *sn*-3 position, that is utilized by DGAT2 to produce the final 2HFA-TAG. Abbreviations not in main text: F.A.S., fatty acid synthesis; G3P; glycerol-3-phosphate.

2HFA-TAG and [$^{12}$C]2HFA-TAG, respectively by lipase treatment and TLC as described above. Acyl-CoAs were chemically synthesized from the free fatty acids and CoA as described by ref. 60. All the substrates were quantified by GC-FID as described below.

## Gene constructs, yeast transformation, and microsomal preparation

Synthetically synthesized *DGAT1*, *DGAT2*, and *DGAT3* genes of Physaria and Arabidopsis and *PfeTAGL1* were cloned individually into the yeast expression vector pYES2/NT-C under the control of the galactose-inducible promoter GAL1. Genes were cloned as N-terminal His6-tag fusions, and the URA3 marker was used for yeast selection. The recombinant plasmids harboring *DGAT1*, *DGAT2*, and *DGAT3* genes were transformed into TAG-deficient H1246 *Saccharomyces cerevisiae* strain[37], whereas the plasmid harboring *PfeTAGL1* is transformed into *Saccharomyces cerevisiae* strain, INVSc1 by using lithium acetate method as in the Invitrogen manual. Microsomal preparations were carried out as described by ref. 21. Total protein content of the microsomal membrane fraction was determined by Pierce BCA Protein Assay kit (Thermo Fisher Scientific) using BSA as standard. Expressed proteins in the microsomal fraction were confirmed by western blotting using anti-6x-His tag monoclonal antibody (HIS.H8, Invitrogen, USA; Cat.no:MA1-21315) produced in the mouse at a dilution of 1:3000 and rabbit anti-mouse IgG secondary antibody tagged with HRP (Invitrogen, USA; Cat.no: 61-6520) at a dilution of 1:6000. Blots were developed with Clarity™ Western ECL substrate (BIO-RAD, USA).

## In vivo metabolic labeling and neutral lipid profiling of Yeast expressing *Pfe*TAGL1

In vivo metabolic labeling of INVSc1 yeast strain over-expressing PfeTAGL1 protein and subsequent neutral lipid profiling was performed as described by[27].

## Overexpression and Purification of *Pfe*TAGL1 protein

A truncated version of *PfeTAGL1* gene that lacks 186 bp at the 5′ end (encoding the hydrophobic *N*-terminus) was amplified from the synthetic PfeTAGL1 gene using primers 5′-ATAT GGATCCATGACA-GATGTTAATGATTTACCTCC-3′ and 5′-ATATCTCGAGTTACTGGTG TTTAGCTTCGTTTG (adding *BamH*I and *Xho*I sites, respectively) and cloned into the same sites in the bacterial expression vector pET28a(+). The recombinant plasmid and empty vector control were transformed into BL21-CodonPlus (DE3)-RIPL cells (Agilent, USA). The cells were induced at 25 °C for 12 h using 0.4 mM IPTG. The recombinant His-tag protein was purified under native conditions using the ProBond™ Purification system (Invitrogen, USA). Briefly, the cell supernatant was loaded onto the Ni-NTA column and the protein was eluted using 250 mM Imidazole according to the manufacturer's instructions. After SDS-PAGE analysis, different elution fractions were pooled, concentrated, and buffer exchanged (50 mM Tris buffer pH 7.5 and 10% glycerol) using Amicon Ultra 15 ml centrifugal units (Millipore Sigma, USA). The purified protein was further confirmed by immunoblotting and quantified using BCA Protein Assay kit as described above.

## In vitro enzymatic assays

DGAT assays were performed as described in[61] with some modifications. The 100 µl reaction volume contained 50 mM HEPES buffer pH 7.2, 5 mM MgCl2, and 1 mg/ml fatty acid-free BSA. Five nmol of acyl-CoA and 20 nmol of [$^{14}$C]rac-diolein or [$^{14}$C]1HFA-rac-DAG were used for substrate specificity assays. For enantiomer specificity assays, 1 nmol of [$^{14}$C]18:1-CoA or 5 nmol of [$^{14}$C]20:1OH-CoA was used as an acyl donor, and 20 nmol or 10 nmol of *sn*-1,2-diolein or 20 nmol *sn*-1,2/2,3-rac-diolein was used as acyl acceptor. Reactions without exogenously added DAG were used as controls to subtract background activity from the presented results. Microsomal membranes corresponding to 50 µg of microsomal protein were used and the reactions were incubated by shaking at 1250 rpm for 20 min then terminated by the addition of 120 µl of 150 mM acetic acid and 500 µl of chloroform:methanol (1:1, v/v). Lipids in the chloroform phase were extracted and an aliquot corresponding to 1/5 of the total extract was taken to determine total radioactivity using liquid scintillation counting on a PerkinElmer Tri-Carb 4910. The remaining chloroform was dried under a nitrogen stream, re-dissolved in 40 µl of chloroform, and the neutral lipids were separated on TLC using the solvent systems specific for non-hydroxy and hydroxy-containing lipids as described above. The labeled neutral lipids were measured on a Typhoon FLA 7000 phosphor imager and the relative amount of radioactivity in TAG was quantified using ImageQuant TL 7.0 (GE healthcare, United States). TAG was quantified using the relative amount of radioactivity as determined by TLC-phosphor imager and the total amount of radioactivity as measured by the LSC.

In vitro lipase activity assays and competitive lipase assays using purified *Pfe*TAGL1 protein were carried out as in[27] except for the use of Britton−Robinson buffer for pH-dependent assays. For competitive lipase assays, an equimolar ratio (1:1) of [$^{14}$C]triolein:[$^{12}$C]triolein, or

[$^{14}$C]triolein:[$^{12}$C]1HFA-TAG, or [$^{14}$C]triolein:[$^{12}$C]2HFA-TAG substrates dissolved in ethanol was used.

## Vector constructs for protein:protein interaction studies

*P. fendleri* genes were commercially synthesized (Integrated DNA Technology, Inc.). For split-ubiquitin-based yeast two-hybrid assay, PfeDGAT1 and PfeDGAT2 were used as baits whereas PfeTAGL1, PfeOBL1, PfeSDP1, PfeDGAT1 and PfeDGAT2 were used as preys. PfeDGAT1 and PfeDGAT2 were cloned into bait vector pBT3C and pBT3C-STE respectively where *C*-terminal half of ubiquitin and LexA transcription factor (Cub-LexA) is fused to the *C*-terminus of the proteins of interest (DUALmembrane split-ubiquitin system, Dualsystems Biotech, Schlieren, Switzerland[62]). PfeDGAT2 was modified to enhance protein expression by *Saccharomyces cerevisiae* codon optimization as suggested by ref. 63. PfeTAGL1, PfeOBL1, PfeSDP1, PfeDGAT1, and PfeDGAT2 were cloned into prey vector pPR3N which has the *N*-terminal half of ubiquitin and an HA epitope fused to the *N*-terminus of the proteins of interest. For protein localization studies PfeDGAT1, PfeDGAT2, PfeTAGL1, PfeOBL1, and PfeSDP1 tagged at their *N*-termini with GFP were made by GATEWAY cloning. First these genes were amplified by PCR to create attB sites and used to make entry clones in the pDONR-207 vector. These entry vectors were further used to make destination vectors pUBN-GFP-Dest (tagging GFP at *N*-terminus of the protein), transformed into *Agrobacterium tumefaciens* GV3101, and eventually into plants. *Agrobacterium* transformation was performed using freeze thaw[64]. Those entry clones were also used to create destination vectors for the BiFC assay. PfeDGAT1 and PfeDGAT2 were cloned into nYFPC1-pSITE vector which has *N*-terminus of YFP tagged at *N*-terminus of the protein. Destination vector cYFPC1-pSITE was used to create *C*-terminus of YFP tagged at *N*-terminus of PfeTAGL1, PfeOBL1, PfeSDP1, PfeDGAT1 and PfeDGAT2 for BiFC assays.

## Split-ubiquitin-based yeast two-hybrid assays

Yeast two-hybrid assays were performed as described as in ref. 65. Bait and prey plasmids were co-transformed into yeast *Saccharomyces cerevisiae* strain NMY51 strain and plated on nonselective synthetic dextrose medium lacking leucine and tryptophan (SD-LT) and incubated at 30 °C for 2–3 days. After culturing overnight in nonselective media to generate the initial cell suspensions, interactions between different protein partners were measured qualitatively by observing growth after 2–3 days at 30 °C of serial fivefold dilutions on plates containing selective media lacking adenine, histidine, leucine, and tryptophan, and quantitatively by measurement of β-galactosidase activity, measured in Miller Units (Yeast beta-galactosidase Assay kit, Thermo Scientific).

## Transient transfection assays and imaging

*Nicotiana benthamiana* wild-type and High Oil lines were grown in growth chamber with ~100 µmol m$^{-2}$ s$^{-1}$ and 16 h day and 8 h night cycle. Four to five weeks old plants were used for agrobacterium infiltration. For protein localization, overnight culture of agrobacterium harboring different GFP-tagged *P. fendleri* genes (1.0 OD$_{600}$) was resuspended in infiltration media [10 mM MES buffer pH 5.6, 10 mM MgCl$_2$·6H$_2$O, 200 µM Acetosyringone] and incubated at room temperature for 3 h. The cultures were mixed 1:1 with agrobacterium harboring viral suppressor protein p19 (0.5 OD$_{600}$) and infiltrated into abaxial side of the leaves. For BiFC assays, agrobacterium harboring constructs with nYFPC-P1(Protein 1), cYFPC-P2 (Protein 2), and p19 were mixed 1:1:1 in infiltration media to make a final OD$_{600}$ of 0.25 for each culture. Leaves of 4–5 weeks old plants were infiltrated with these agrobacterium constructs, kept in dark overnight before moving them to optimal growth light condition and observed under confocal microscope after 2 days. Oil bodies in leaf pavement cells of High Oil *N. benthamiana* plants were stained for 30 min with 2 µg/ml Nile Blue A

(Sigma-Aldrich) freshly diluted in 50 mM PIPES buffer (pH 7.0) from 1 mg/ml stock in DMSO.

Imaging was performed using a Leica SP8 or SP8X confocal laser scanning microscope equipped with a 40× NA1.3 oil immersion objective. GFP was excited with 488 nm and emitted light was acquired between 500–540 nm. GFP and Nile Blue A were excited by a 489 nm and emission was acquired between 500–540 nm and 560–620 nm for GFP and Nile Blue A respectively. Chlorophyll autofluorescence was collected between 640-720 nm. For GFP/mCherry co-localization, samples were excited at 489 nm and 552 nm and the emitted light was collected between 500–540 nm for GFP and 597–620 nm for mCherry. Leaf sections infiltrated with BiFC constructs were excited at 514 nm for EYFP and emitted light was recorded between 525-590 nm. All images are Z-stack projection images.

## RNAi constructs preparation and creation of transgenic knockdown (KD) lines

*To prepare* RNAi constructs for PfeDGAT1, PfeDGAT2, and PfeTAGLL1, target gene fragments between 180-200 bp were PCR amplified using respective chemically synthesized full-length genes (IDT) as a template with Phusion Taq (New England Biolabs). For all three-target gene fragments the forward arm was cloned at *Not*I/*Xma*I, whereas the reverse arm was cloned at *Pst*I/*Sac*II sites of the J2 vector under seed-specific 2S albumin promoter (Supplementary Fig. S7)[66]. These clones were sequence-verified, and the promoter: gene: terminator cassette was released by digestion with *Asc*I and cloned into binary vector pB9[66]. A complete list of the primers used for qRT-PCR is presented in Supplementary Table S1.

Plant transformation was performed using the Agrobacterium GV3101 strain carrying either *Pfe*DGAT1-RNAi or *Pfe*DGAT2-RNAi or *Pfe*TAGL1-RNAi construct in binary vector pB9[66]. The in vitro *P. fendleri* grown plant's mature leaves were used for the transformation. Tissue culture and transformation protocol followed as described previously[67] with some modifications, using Basta 1.2 mg/L for transgenic selection as described in[27].

## RNA isolation and quantitative Real-Time PCR (qRT-PCR) analysis

Total RNA was extracted from the seeds of the selected knockdown lines using a Trizol reagent. Total RNA was treated with RNase-free DNase (Qiagen) and cleaned using an RNeasy mini kit (Qiagen). RNA concentration and purity were quantified using a NanoDrop 2000 spectrophotometer (Thermo Scientific). One microgram of Total RNA was used to generate cDNA using iScript cDNA synthesis kit (Bio-Rad). qRT-PCR analysis was carried out with CFX96 Real-Time system (Bio-rad) using Power SYBR Green PCR Master Mix (Applied Biosystems). The relative expression was calculated using 2$^{-(\Delta\Delta CT)}$ method. 18S gene was used as an internal reference gene and the Physaria_fen32 control plant seed sample was used as a calibrator.

## Plant growth conditions for transgenic A. thaliana lines

All *Arabidopsis thaliana* plant lines were grown in a growth chamber set at 22 °C with a light intensity of ~150–200 µmol m$^{-2}$ s$^{-1}$ and 16-hour day and 8-hour night photoperiod. Arabidopsis seeds were first surface sterilized with 70% ethanol, 10% bleach with 0.1% sodium dodecyl sulfate (SDS), sterile water, and plated on Murashige and Skoog media supplemented with 1% sucrose. After that, plates were vernalized for 3 days at 4 °C and transferred under light for 2 weeks, before transplanting to soil. Plants were watered twice a week and provided with fertilizer solution NPK 20-20-20 (0.9957 gL$^{-1}$) once a week. All the solvents were HPLC grade or above and chemicals were from Fisher Scientific unless indicated.

## Cloning and creating transgenic *A. thaliana* lines

*PfeDGAT1, PfeDGAT2, PfeTAGL1*, and *PfeSDP1* were amplified by PCR using Phusion polymerase to create 5′ *Not*I and 3′ *Sac*II or *Sal*I restriction sites (in the case of PfeSDP1), which were digested with respective enzymes before ligation. All enzymes were from New England Biolabs. and all plasmids were constructed using plant expression vectors from ref. 66. To make shuttle plasmids, ORFs of *PfeDGAT1* and *PfeDGAT2* were cloned into pB34 with 2S albumin promoter and *glycinin* terminator whereas *PfeTAGL1* and *PfeSDP1* were cloned into pB35 with *beta-conglycinin* promoter and soy *oleosin* terminator. The promoter:gene:terminator cassettes for each of these genes were then put into binary vector pB110, which contains a *DsRed* selectable marker (Supplementary Fig. S11). The integrity of all the plasmids were confirmed by sequencing as well as digestion of the plasmid with restriction enzymes before using for agrobacterium and plant transformation. *A. tumefaciens* strain GV3101 was transformed with these different binary vectors using a freeze-thaw method[64] and selected on Rifampicin/Gentamycin/Kanamycin plates and the presence of genes was confirmed by colony PCR. *Arabidopsis thaliana* line (RcFAH) expresses *RcFAH12* and produces hydroxy fatty acids, and was generated by a cross of wild-type with CL37 (which has *RcFAH12* in the *fae1* mutant background[68]) to produce 20:1OH fatty acids. Agrobacterium-based transformation of Arabidopsis was through the floral dip method[69]. Transformed plants were grown to maturity, seeds were harvested and successful transformants were selected based on DsRed fluorescence when observed under green light with red filter, red T1 seeds were moved to next generation. Oil content and fatty acid composition of T2 generation was analyzed as mentioned in the section below and top six lines were moved to next generation to analyze for homozygosity.

## Oil content and fatty acid composition analysis

Two to three milligrams seeds and 25 µg of 17:0 TAG internal standard were converted into FAMEs using 5% $H_2SO_4$ in methanol and heated at 80–85 °C for 1.5 h. FAMEs were extracted with 1 ml hexanes and 1.5 ml 0.88% potassium chloride solution. After centrifugation at $1000 \times g$ for 2 min, the hexane layer was transferred to GC vial and analyzed by an Agilent 7890 GC-FID, with data collection by Open Lab Chemstation edition for GC systems version 38 (30-May-2018) C.01.09[27].

## Lipid extraction, separation of different lipid species, and quantification

One hundred milligram seeds from T4 lines were quenched in isopropanol with 0.01% w/v butylated hydroxytoluene at 85 °C for 10 min and total lipid was extracted using hexanes-isopropanol[70]. 1 mg lipid was separated into different HFA-TAG molecular species by HPLC[59,71], collected fractions were converted to FAME and fatty acid composition determined by GC-FID as above.

## Statistics and reproducibility

All the results were expressed as mean ± SD or mean ± SEM and statistical significance was determined in GraphPad Prism Version 10.1.2 using either one-way or two-way ANOVA or multiple *t*-tests based on the number of datasets being compared. *P* values < 0.05 were considered statistically significant. All the information related to the statistical methods, sample size and their entity and exact *p* value were presented in the respective figure legends and source data file. "No data were excluded from the analyses". "No statistical method was used to predetermine sample size. The Investigators were not blinded to allocation during experiments and outcome assessment".

## Reporting summary

Further information on research design is available in the Nature Portfolio Reporting Summary linked to this article.

## Data availability

Data supporting the findings of this work are available within the paper and its supplementary information and source data files. A reporting summary for this article is available as a supplementary information file. The datasets and plant materials generated and analyzed during the current study are available from the corresponding author upon request. The nucleotide sequence of genes used in this study are available from the GenBank under the accession numbers: PP588719 – PfeDGAT1; PP588720 – PfeDGAT2; PP588721- PfeTAGL1. Source data are provided with this paper.

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

## Acknowledgements

This material is based upon work supported by the United States Department of Agriculture National Institute of Food and Agriculture grants #2020-67013-30899 to P.B. and G.C., #2023-67013-39022 to P.B. and A.A., and the Hatch Umbrella Project #1015621 to P.B.; the National Science Foundation (MCB #2242822 to P.B., J.S., and A.S.); and the U.S. Department of Energy, Office of Science, Office of Biological and Environmental Research, under award number DE-SC0023142 to P.B. Full-length sequences for all *P. fendleri* genes used in this study were kindly provided by Dr. Patrick Horn of the University of North Texas.

*Nicotiana benthamiana* High Oil lines were kindly provided by Dr. Xue-Rong Zhou, CSIRO, Australia.

## Author contributions

P.B. conceived the project, procured funding (along with A.A., G.C., J.S. and A.S.), agreed to serve as the author responsible for contact, and ensured communication. P.P., S.B., A.A., G.C., J.S., A.S. and P.B. designed experiments, analyzed the data, and prepared figures. P.P., S.B., A.A., K.J. and G.C. performed the experiments. All authors contributed to writing the article and approved the submitted version.

## Competing interests

The authors declare no competing interests.
