## [Peer Review File · Nature Communications]

REVIEWER COMMENTS

Reviewer #1 (Remarks to the Author):

This is an outstanding paper providing a detailed analysis of three key enzymes needed for the production of hydroxy fatty acid containing TAG from *P. fendleri* constituting a newly proposed metabolon. The paper addresses a long-standing problem in the field and identifies key players and mechanisms providing substantial novelty. The characterization of the activities of the two involved acyltransferases and the respective TAG lipase are expertly done. These are not easy enzymes to work with and the generation of specific substrates for the specificity assays is impressive and the assays are well done. I also do not see any issues with the localization experiments or the interaction studies. I think they are well controlled, and the data are conclusive. The authors also take their work one step further in trying to assess the efficacy of the three enzymes for the engineering of HFA containing TAG by generating and analyzing the respective *Arabidopsis* lines. The results of these complementary experiments also support the results of the biochemical analyses. It would interest me to know how close these results (amount of HFA) are to the *P. fendleri* oil. Maybe this could be mentioned unless I missed it.

My suggestions for improvements are only minor and I have made comments in the enclosed Manuscript PDF.

1. There is confusion about expressing genes versus proteins. All genes are expressed and this should be consistent throughout.
2. The introduction is fairly lengthy and somewhat repeated in the Discussion. As such, the paper could benefit from from condensing of the text.
3. In Figure 5C the TAG in the R mehei lane the TAG band runs differently. This should be explained.
4. It took me a while to understand the competition assay in Figure 6. This could be explained better. The description is not very intuitive as is.
5. In the same figure there needs to be a % reduction in the legend, if I am not mistaken.

Reviewer #2 (Remarks to the Author):

Physaria fendleri accumulates triacylglycerol molecules in its seeds that have an 'unusual' very long chain hydroxylated fatty acid (lesquerolic acid, LA) esterified to the outer (sn-1&3) positions on the glycerol backbone. The authors have previously found, using an in vivo isotopic labelling approach, that developing *P. fendleri* seeds initially synthesize TAG with LA at the sn-3 position and then over time TAG gradually accumulates containing LA at both sn-1 and 3 positions. They proposed a TAG remodelling mechanism to explain their data where the common fatty acid at sn-1 is removed from TAG with LA at sn-3 by a lipase and the second LA group is esterified to the sn-2,3-DAG by a DAG acyltransferase (DGAT).

In this manuscript the authors present biochemical and molecular data to support this pathway model. They show that *P. fendleri* DGAT1 has an in vitro substrate specificity that's most appropriate for the first acylation of sn-1,2-DAG with LA and DGAT2 with the second acylation of sn-2,3-DAG carrying LA at the sn-1 position. They also identify a lipase (TAGL1) that in vitro has preference for removing normal fatty acids (oleic acid) from the sn-1 positions over LA. They show that all three proteins are ER localised when expressed as GFP fusions in *N. benthamiana* leaves, but TAGL1 also localises to lipid droplets and associates with DGAT1 (based on yeast two-hybrid and BiFC assays). The authors show that overexpression of PfeDGAT1, PfeDGAT2 and PfeTAGL1 in *Arabidopsis* expressing the castor hydroxylase RcFAH increases seed lipid content and hydroxy fatty acid content (at least in case of PfeDGAT1 and PfeTAGL1).

I think this is an important manuscript in the field that presents interesting findings that support an exciting hypothesis that is transformative for our understanding of how storage lipids are synthesized in some seeds. The manuscript is quite long, but it's very well written, methods appear sound and the data appear to be of high quality, and generally do support the conclusions.

My only significant criticism is that the data provide lines of corroborative evidence for the authors' hypothesis rather than testing it directly. To test whether TAGL1 and DGAT2 are necessary for production of TAG with two LA moieties, and DGAT1 one, ideally the gene(s) should be knocked out (or down) in *P. fendleri* and the effect on TAG composition analysed. Admittedly, *P. fendleri* is not an easy plant to transform, but the literature suggests it is possible and the authors have published work on *P. fendleri* RNAi lines recently where they performed a very similar analysis (<https://doi.org/10.3389/fpls.2022.931310>).

The authors have performed heterologous gain-of-function experiments in *Arabidopsis* expressing RcFAH. I think these experiments are useful, but are limited to showing what PfeDGAT1, PfeDGAT2 and PfeTAGL1 can do rather than what they do do. The expression of PfeDGAT1, PfeDGAT2 and PfeTAGL1 each changes lipid composition. It is interesting that PfeTAGL1 expression increases hydroxy FA content, but I am not sure that the changes amount to clear evidence that PfeDGAT1, PfeDGAT2 and PfeTAGL1

operate according to the authors' model. As the authors say in the manuscript, *Arabidopsis* expressing RcFAH mainly produces ricinoleic acid that accumulates at sn-2 in TAG while *P. fendleri* mainly produces LA that accumulates at sn-1&3.

Minor comments

L58 – 'phosphatidylcholine (PC), which is the site for fatty acid desaturation in all plants' suggest rewording because non-PC desaturation is occurring in plastids.

L98 – to synthesize

L256 - In Fig 5C TAGL1 appears to be producing MAG as well as DAG. I don't think this is unusual for a TAG lipase, but how does it fit with the authors' hypothesis? It suggests to me that TAGL1 might be sn-1/3 specific or have no regioselectivity. The manuscript reads as though TAGL1 might be a sn-1 specific lipase as well as having an acyl preference against LA, but the data suggest that TAGL1 may remove normal fatty acids from the sn-1 and/or sn-3 position of TAGs that happen to have normal fatty acids esterified there. Could this be happening in vivo and perhaps help enrich LA at sn-1 as well as sn-3?

L285 – 'and total HFA content (Figure 7B,C)'. The lack of an asterisk on the Fig suggests PfdDGAT2 expression doesn't increase total HFA content.

L401 – PC-derived DAG

Response to reviewers is italicized in blue below.

REVIEWER COMMENTS

Reviewer #1 (Remarks to the Author):

This is an outstanding paper providing a detailed analysis of three key enzymes needed for the production of hydroxy fatty acid containing TAG from *P. fendleri* constituting a newly proposed metabolon. The paper addresses a long-standing problem in the field and identifies key players and mechanisms providing substantial novelty. The characterization of the activities of the two involved acyltransferases and the respective TAG lipase are expertly done. These are not easy enzymes to work with and the generation of specific substrates for the specificity assays is impressive and the assays are well done. I also do not see any issues with the localization experiments or the interaction studies. I think they are well controlled, and the data are conclusive. The authors also take their work one step further in trying to assess the efficacy of the three enzymes for the engineering of HFA containing TAG by generating and analyzing the respective *Arabidopsis* lines. The results of these complementary experiments also support the results of the biochemical analyses. It would interest me to know how close these results (amount of HFA) are to the *P. fendleri* oil. Maybe this could be mentioned unless I missed it.

*Thank you for recognizing the value of the manuscript and the hard work included. As for the comparison of *P. fendleri* oil to *Arabidopsis*, we apologize for this omission. We now indicate in the introduction that HFA accumulate to 60% of *P. fendleri* seed oil.*

*Even though the individual TAG remodeling genes increase HFA in *Arabidopsis*, the lower total amounts in the engineered *Arabidopsis* lines are likely due to multiple factors including competition from endogenous *Arabidopsis* genes to make TAG without HFA, the fact that *Arabidopsis* mostly makes ricinoleic acid vs lesquerolic acid which we demonstrate is not the best HFA substrate for *P. fendleri* acyltransferases (Fig. 1), and likely the need to combine the knockdown of endogenous genes with multiple stacked HFA-selective acyltransferases to greatly increase *Arabidopsis* HFA production.*

*Nevertheless, we clearly demonstrate that each of the *P. fendleri* TAG remodeling genes can enhance HFA accumulation in a different plant species. Which, as stated by the reviewer, takes the work a step farther and indicates that engineering of TAG remodeling may be a new valuable approach to control oilseed fatty acid compositions.*

My suggestions for improvements are only minor and I have made comments in the enclosed Manuscript PDF.

1. There is confusion about expressing genes versus proteins. All genes are expressed and this should be consistent throughout.

Thank you for pointing this out, we have gone through the manuscript making changes to clarify genes vs proteins.

2. The introduction is fairly lengthy and somewhat repeated in the Discussion. As such, the paper could benefit from from condensing of the text.

We agree there is some unnecessary repetition, therefore we have removed those parts from the discussion.

3. In Figure 5C the TAG in the R mehei lane the TAG band runs differently. This should be explained.

The commercial preparation of R. mehei TAG lipase contains some extra lipophilic compounds (probably detergents or emulsifiers) that comigrate near TAG, and push the TAG band up a little. These are not ¹⁴C labeled and thus do not affect quantitation, only the migration of TAG in that lane. Considering that is a control lane that serves only to demonstrate the degradation of ¹⁴C-TAG to produce ¹⁴C free fatty acids and DAG (the migration of which are not affected), this does not affect the results or conclusions of the manuscript.

4. It took me a while to understand the competition assay in Figure 6. This could be explained better. The description is not very intuitive as is.

Thank you for pointing this out, we have added a description of how the assay works to the Figure 6 legend.

5. In the same figure there needs to be a % reduction in the legend, if I am not mistaken.

The figure now contains % in both the figure and the legend.

Reviewer #2 (Remarks to the Author):

Physaria fendleri accumulates triacylglycerol molecules in its seeds that have an 'unusual' very long chain hydroxylated fatty acid (lesquerolic acid, LA) esterified to the outer (sn-1&3) positions on the glycerol backbone. The authors have previously found, using an in vivo isotopic labelling approach, that developing P. fendleri seeds initially synthesize TAG with LA at the sn-3 position and then over time TAG gradually accumulates containing LA at both sn-1 and 3 positions. They proposed a TAG remodelling mechanism to explain their data where the common fatty acid at sn-1 is removed from TAG with LA at sn-3 by a lipase and the second LA group is esterified to the sn-2,3-DAG by a DAG acyltransferase (DGAT).

In this manuscript the authors present biochemical and molecular data to support this pathway model. They show that P. fendleri DGAT1 has an in vitro substrate specificity that's most appropriate for the first acylation of sn-1,2-DAG with LA and DGAT2 with the second acylation of sn-2,3-DAG carrying LA at the sn-1 position. They also identify a lipase (TAGL1) that in vitro has preference for removing normal fatty acids (oleic acid) from the sn-1 positions over LA. They show that all three proteins are ER localised when expressed as GFP fusions in N. benthamiana leaves, but TAGL1 also localises to lipid droplets and associates with DGAT1 (based on yeast two-hybrid and BiFC assays). The authors show that overexpression of PfdGAT1, PfdGAT2 and PfdTAGL1 in Arabidopsis expressing the castor hydroxylase RcFAH increases seed lipid content and hydroxy fatty acid content (at least in case of PfdGAT1 and PfdTAGL1).

I think this is an important manuscript in the field that presents interesting findings that support an exciting hypothesis that is transformative for our understanding of how storage lipids are synthesized in

some seeds. The manuscript is quite long, but it's very well written, methods appear sound and the data appear to be of high quality, and generally do support the conclusions.

My only significant criticism is that the data provide lines of corroborative evidence for the authors' hypothesis rather than testing it directly. To test whether TAGL1 and DGAT2 are necessary for production of TAG with two LA moieties, and DGAT1 one, ideally the gene(s) should be knocked out (or down) in *P. fendleri* and the effect on TAG composition analysed. Admittedly, *P. fendleri* is not an easy plant to transform, but the literature suggests it is possible and the authors have published work on *P. fendleri* RNAi lines recently where they performed a very similar analysis (<https://doi.org/10.3389/fpls.2022.931310>).

The authors have performed heterologous gain-of-function experiments in *Arabidopsis* expressing RcFAH. I think these experiments are useful, but are limited to showing what PfeDGAT1, PfeDGAT2 and PfeTAGL1 can do rather than what they do do. The expression of PfeDGAT1, PfeDGAT2 and PfeTAGL1 each changes lipid composition. It is interesting that PfeTAGL1 expression increases hydroxy FA content, but I am not sure that the changes amount to clear evidence that PfeDGAT1, PfeDGAT2 and PfeTAGL1 operate according to the authors' model. As the authors say in the manuscript, *Arabidopsis* expressing RcFAH mainly produces ricinoleic acid that accumulates at sn-2 in TAG while *P. fendleri* mainly produces LA that accumulates at sn-1&3.

The reviewer is correct that our in vitro work is consistent with our TAG remodeling hypothesis, but lacks direct in vivo evidence, and the interpretation of the Arabidopsis gain-of-function experiments is complicated by combined endogenous Arabidopsis metabolism with the expressed transgenes. As mentioned to Reviewer 1, fully converting Arabidopsis TAG to Physaria TAG will likely require extensive modification of endogenous Arabidopsis metabolism as well as introduction of TAG remodeling genes from P. fendleri. However, the addition of in vivo P. fendleri genetic evidence to our prior in vitro results and gain-of-function demonstrations of TAG remodeling gene functions would greatly enhance confirmation of the P. fendleri TAG remodeling mechanism.

Transformation of P. fendleri is possible, but not easy or fast. Tissue culture-based transformation of P. fendleri suffers from issues of chimeric tissues and due to inherent self-incompatibility of P. fendleri the propagation and production of seeds requires laborious hand self-pollination at immature flower stages. In our hands production of a homozygous P. fendleri line takes nearly two years. Despite these challenges, we were already working on producing seed specific RNAi knockdown lines of DGAT1, DGAT2, and TAGL1 that we were planning on using for a follow-up manuscript. However, we are happy to include that data in this manuscript.

The new Figure 7 clearly demonstrates the importance for each gene in P. fendleri total oil accumulation and fatty acid composition. In addition, through qPCR we investigated if the knockdown of individual genes may be compensated for by changes in expression of any of the other TAG remodeling genes (or PDAT1 which is known to compensate for the dgat1 mutant in Arabidopsis). The gene expression changes discussed in the new results section help to explain the fatty acid composition changes of the knockdown lines, and are consistent with our enzyme assays and the utilization of different isomers of DAG by DGAT1 and DGAT2. Therefore, we believe our in vivo data confirms the conclusions from our prior in vitro and gain-of-function experiments regarding the in vivo roles of DGAT1, DGAT2 and TAGL1 for P. fendleri TAG remodeling.

Minor comments

L58 – ‘phosphatidylcholine (PC), which is the site for fatty acid desaturation in all plants’ suggest rewording because non-PC desaturation is occurring in plastids.

Thank you for pointing this out. We reworded it to “...the extra-plastidial site for fatty acid desaturation....”

L98 – to synthesize

Fixed

L256 – In Fig 5C TAGL1 appears to be producing MAG as well as DAG. I don’t think this is unusual for a TAG lipase, but how does it fit with the authors’ hypothesis? It suggests to me that TAGL1 might be sn-1/3 specific or have no regioselectivity. The manuscript reads as though TAGL1 might be a sn-1 specific lipase as well as having an acyl preference against LA, but the data suggest that TAGL1 may remove normal fatty acids from the sn-1 and/or sn-3 position of TAGs that happen to have normal fatty acids esterified there. Could this be happening in vivo and perhaps help enrich LA at sn-1 as well as sn-3?

In the original isotopic labeling publication, it was clear that both normal TAG and 1HFA-TAG could be remodeled. However, 1HFA-TAG was produced and remodeled at much higher levels than normal TAG. Additionally, no labeled MAG was detected from in vivo labeling. Therefore, while TAGL1 probably is also responsible for the removal of common fatty acids from both sn-1 and sn-3 of normal TAG, in vivo evidence for MAG production is lacking. Rapid utilization of the in vivo produced DAG by the DGATs may help to prevent further degradation to MAG. Thus, the major activity from the in vivo isotopic tracing appears to be the removal of the common fatty acids from sn-1 position of 1HFA-TAG and thus this is our major conclusion.

However, in engineered systems that contain a lot of normal TAG, PfeTAGL1 may have a larger role for remodeling both TAG positions. We have now indicated this possibility in the results on the PfeTAGL1 Arabidopsis line.

L285 – ‘and total HFA content (Figure 7B,C)’. The lack of an asterisk on the Fig suggests PfeDGAT2 expression doesn’t increase total HFA content.

In the Arabidopsis PfeDGAT2 line total seed lipid content significantly increases by ~28.3 % and the average mol % HFA composition increases by 4.5% but not significantly when compared to the control line. This is mostly due to a low n value. If we analyze all the PfeDGAT2 lines together, the average HFA content is significantly increased with a p-value of 0.0175 (Supplemental Figure S12).

When total oil and fatty acid composition are combined it is clear that the seeds have an overall increase in total mass of HFA. We added a new supplemental figure S12 which shows an average HFA content increase of 33.6% on μg HFA/mg seed basis for PfeDGAT2 lines.

Additionally, Figure 9C clearly demonstrates that PfeDGAT2 produced a significant increase in 20:1OH in 2HFA-TAG which is consistent with its in vitro selectivity for 20:1OH over 18:1OH. Thus, the gain-of-function experiments clearly demonstrate the ability of PfeDGAT2 to increase HFA content in engineered systems.

L401 – PC-derived DAG

Fixed

REVIEWERS' COMMENTS

Reviewer #1 (Remarks to the Author):

This is an excellent paper with important new results. The authors properly addressed my previous concerns and the inclusion of the RNA1 data further improves the manuscript. I have no further questions.

Reviewer #2 (Remarks to the Author):

Having read the authors response to the reviewers comments and the revised manuscript, which contains further data, I have zero criticisms. This is a truly excellent piece of work. Thank you to the authors for making this important contribution to the field.